# Land Subsidence Susceptibility Mapping in Ca Mau Province, Vietnam, Using Boosting Models

Anh Van Tran [1,2,*], Maria Antonia Brovelli [3], Khien Trung Ha [4], Dong Thanh Khuc [4], Duong Nhat Tran [5], Hanh Hong Tran [1] and Nghi Thanh Le [1]

[1] Faculty of Geomatics and Land Administration, Hanoi University of Mining and Geology, 18 Vien Street, Hanoi 100000, Vietnam; tranhonghanh@humg.edu.vn (H.H.T.); lethanhnghi@humg.edu.vn (N.T.L.)

[2] Geomatics in Earth Sciences (GES), Hanoi University of Mining and Geology, 18 Vien Street, Bac Tu Liem, Hanoi 100000, Vietnam

[3] Department of Civil and Environmental Engineering (DICA) Politecnico di Milano, Piazza Leonardo da Vinci 32, 20133 Milan, Italy; maria.brovelli@polimi.it

[4] Faculty of Bridges and Roads, Hanoi University of Civil Engineering, 55 Giai Phong Street, Hai Ba Trung, Hanoi 100000, Vietnam; khienht@huce.edu.vn (K.T.H.); dongkt@huce.edu.vn (D.T.K.)

[5] Space and Applications Department, University of Science and Technology of Hanoi, 188 Hoang Quoc Viet, Cau Giay, Hanoi 100000, Vietnam

\* Correspondence: tranvananh@humg.edu.vn

**Abstract:** The Ca Mau Peninsula, situated in the Mekong Delta of Vietnam, features low-lying terrain. In addition to the challenges posed by climate change, land subsidence in the area is exacerbated by the overexploitation of groundwater and intensive agricultural practices. In this study, we assessed the land subsidence susceptibility in the Ca Mau Peninsula utilizing three boosting machine learning models: AdaBoost, Gradient Boosting, and Extreme Gradient Boosting (XGB). Eight key factors were identified as the most influential in land subsidence within Ca Mau: land cover (LULC), groundwater depth, digital terrain model (DTM), normalized vegetation index (NDVI), geology, soil composition, distance to roads, and distance to rivers and streams. The dataset includes 2011 points referenced from the Persistent Scattering SAR Interferometry (PSI) method, of which 1011 points are subsidence points and the remaining are non-subsidence points. The sample points were split, with 70% allocated to the training set and 30% to the testing set. Following computation and execution, the three models underwent evaluation for accuracy using statistical metrics such as the receiver operating characteristic (ROC) curve, area under the curve (AUC), specificity, sensitivity, and overall accuracy (ACC). The research findings revealed that the XGB model exhibited the highest accuracy, achieving an AUC and ACC above 0.88 for both the training and test sets. Consequently, XGB was chosen to construct a land subsidence susceptibility map for the Ca Mau Peninsula. In addition, 31 subsidence points measured by leveling surveys between 2005 and 2020, provided by the Department of Survey, Mapping and Geographic Information Vietnam, were used for validating the land subsidence susceptibility from the XGB method. The findings indicate a 70.9% accuracy rate in predicting subsidence susceptibility compared to the leveling measurement points.

**Keywords:** AdaBoost; Gradient Boosting; XGBoost; Ca Mau; subsidence susceptibility

## 1. Introduction

Land subsidence is a common phenomenon in many regions around the world, stemming from a multitude of factors, including both natural processes and human activities. The causes of human-induced land subsidence commonly include groundwater exploitation, mineral extraction, oil and gas extraction, and some others. As land subsidence can lead to geological, hydrogeological, environmental, and/or economic impacts, it garners significant attention from governments, communities, and scientists. While land subsidence may not be entirely avoidable in industries like mining, the sustainable control of

land subsidence can be achieved through government regulations, industrial exploitation monitoring, and rational planning with the aid of predictive subsidence hazard maps [1]. Hence, the role of subsidence hazard maps is immensely crucial, enabling managers to develop mineral extraction, groundwater usage, urban development planning, and land use conversion efficiently.

In recent years, the integration of artificial intelligence and machine learning with geospatial data has gained traction in the field of cartography. Many applications involving machine learning have emerged for constructing predictive models aimed at assessing land subsidence susceptibility.

The first study we would like to present is a study conducted by Rahmati, who utilized two machine learning algorithms, namely MaxEnt (maximum entropy) and GARP (genetic algorithm rule-set production), to construct a subsidence assessment model in Kashmar, Iran [2]. The model incorporated data related to land use, geology, distances to groundwater extraction sites, distances to reforestation projects, distances to fault lines, and groundwater level reduction. The research results indicated that the GARP algorithm outperformed the MaxEnt algorithm in terms of performance and accuracy. Both algorithms yielded reliable subsidence prediction outcomes.

Another study by Abdollahi published the results of utilizing a support vector machine (SVM) model to create a subsidence susceptibility map for Kerman Province, Iran [3]. Data including slope, aspect, elevation, cross-sectional curvature, plan curvature, topographic wetness index (TWI), distance to rivers, groundwater level, geology, pressure variation, land use, and normalized difference vegetation index (NDVI) were incorporated into the model construction. The model yielded results with good accuracy, exhibiting (area under the curve (AUC)) values ranging from 0.857 to 0.894.

In another study [4], the authors established a subsidence susceptibility map in Jakarta, Indonesia; the accuracy of subsidence prediction in Jakarta was assessed using machine learning models, including logistic regression, multilayer perceptron, meta-ensemble AdaBoost, and LogitBoost. They utilized Sentinel-1 (SAR) data from 2017 to 2020 to generate a subsidence-sensitive map. Receiver operating characteristic (ROC) analysis demonstrated that although all the results are relatively close to each other, the AdaBoost algorithm exhibited the highest predictive accuracy (81.1%) compared to the other algorithms, for which predictive accuracy ranged from 79.1% to 80%.

The XGBoost machine learning method was employed by Liyuan Shi and colleagues to develop a subsidence prediction model for the North China Plain region [5]. Factors incorporated into the model included groundwater level variations, the thickness of the Quaternary sediments, and the built-up index (IBI), in combination with Sentinel-1 image-derived subsidence and Persistent Scatterer Interferometry (PSI) measurements. The research results highlighted the excellent accuracy of this method (0.9431).

A published study conducted by Elham Rafiei Sardooi compared four machine learning and statistical models, namely the evidential belief function (EBF), Index of Entropy (IoE), support vector machine (SVM), and random forest (RF), for subsidence prediction in the Rafsanjan plain region of Iran [6]. The model training data included 11 factors, namely slope, aspect, topographic wetness index (TWI), plan and profile curvatures, normalized difference vegetation index (NDVI), land use, lithology, distance to rivers, groundwater drawdown, and elevation. The study utilized the Boruta algorithm to determine the significance of the causal factors. The research findings revealed that the SVM model achieved the highest predictive accuracy (AUC = 0.967, TSS = 0.91), followed by RF (AUC = 0.936, (True Skill Statistic—TSS = 0.87), EBF (AUC = 0.907, TSS = 0.83), and IoE (AUC = 0.88, TSS = 0.8). A comprehensive study conducted by Bui involved the comparison of machine learning techniques, including Bayesian logistic regression, support vector machine (SVM), logistic model tree, and alternative decision tree models, to construct a land subsidence risk prediction model in South Korea [7]. Nevertheless, as far as our understanding goes, the study might be subject to significant bias due to a limited number of sample points used for training and validation.

Wang and colleagues have published a study on the application of land subsidence prediction using the artificial neural network BPNN and the random forest (RF) method in the Shandong region of China [8]. The data used for subsidence prediction consisted of groundwater level variations and subsidence data from the period 2017 to 2020, identified through the SBAS-InSAR technique. The research results indicated that the BPNN model exhibited higher accuracy than the RF model.

Mohammadifar applied stacking- and voting-based ensemble deep learning models (SEDL and VEDL) along with active learning (AL) to establish subsidence susceptibility maps in the Minab and Shamil–Nian plains of Hormozgan Province, southern Iran [9]. According to the study, groundwater level decline had a significant impact on the models' output results. Based on Taylor diagrams and $R^2$ values (model performance assessment indicator), the predictive outcomes of the SEDL-AL model ($R^2 > 95\%$) demonstrated higher performance and accuracy compared to the SEDL model.

With a diverse range of machine learning algorithms mentioned above applied in many different countries, each region has distinct geographical and geological features. Models are not completely effective for every area; they need to be tailored to the specific geographical features of the study area. In this paper, we aim to explore several boosting machine learning algorithms—Adaboost, Gradient Boosting, and XGBoost—to predict land subsidence susceptibility in Ca Mau Peninsula, Vietnam. Located in the southernmost part of Vietnam, Ca Mau, characterized by its low terrain, is vulnerable to a multitude of environmental threats including land subsidence, rising sea levels, flooding, and saltwater intrusion. Research by Erban demonstrated subsidence in the Ca Mau Peninsula and across the entire Mekong Delta in excess of several centimeters per year, surpassing the present absolute sea level rise [10].

The reason for choosing the boosting method is due to the flat terrain and low topography in this delta where the main cause of land subsidence is still unknown. Boosting models, based on decision trees, merge weak models to form a strong model. The weights of the next layers are updated from the previous weights, which can help improve the accuracy of the prediction. The sample data input includes land subsidence points determined by the Persistent Scattering SAR Interferometry (PSI) method and leveling survey. Moreover, the Ca Mau area lacks any prior study utilizing boosting models for land subsidence prediction, but most studies in this area have focused on monitoring subsidence. Hence, our experiment can be considered pioneering, aiding in effective and sustainable land use planning in this region.

## 2. Study Area

Ca Mau Province is located in the southernmost part of the Mekong Delta, Vietnam, encompassing both mainland and several islands, with a total area of 5329 square kilometers, equivalent to about 13.10% of the Mekong Delta's area and 1.57% of the country's total area. It shares its northern border with Kiên Giang Province, its eastern border with Bac Lieu Province, its western border with the West Sea (Gulf of Thailand), and its southern and eastern borders with the East Sea. Figure 1 shows the location of Ca Mau province on the Vietnam map.

### 2.1. Topographical and Soil Characteristics

Ca Mau is situated in a region bordering both the East Sea and the West Sea (Gulf of Thailand), with land that originates from sedimentary processes, featuring relatively low and fairly flat terrain. The average ground elevation is approximately 0.6 m. Low-lying areas have elevations around 0.2 m, while higher ground reaches elevations of about 0.8–1.5 m. Most of the land lies below the high tide water level, making it susceptible to flooding, particularly during high tides [11].

Ca Mau is a newly formed land area created by sedimentation, comprising marine sediments and river sediments. These types of land generally have a detrimental impact on both surface water quality and the province's groundwater source.

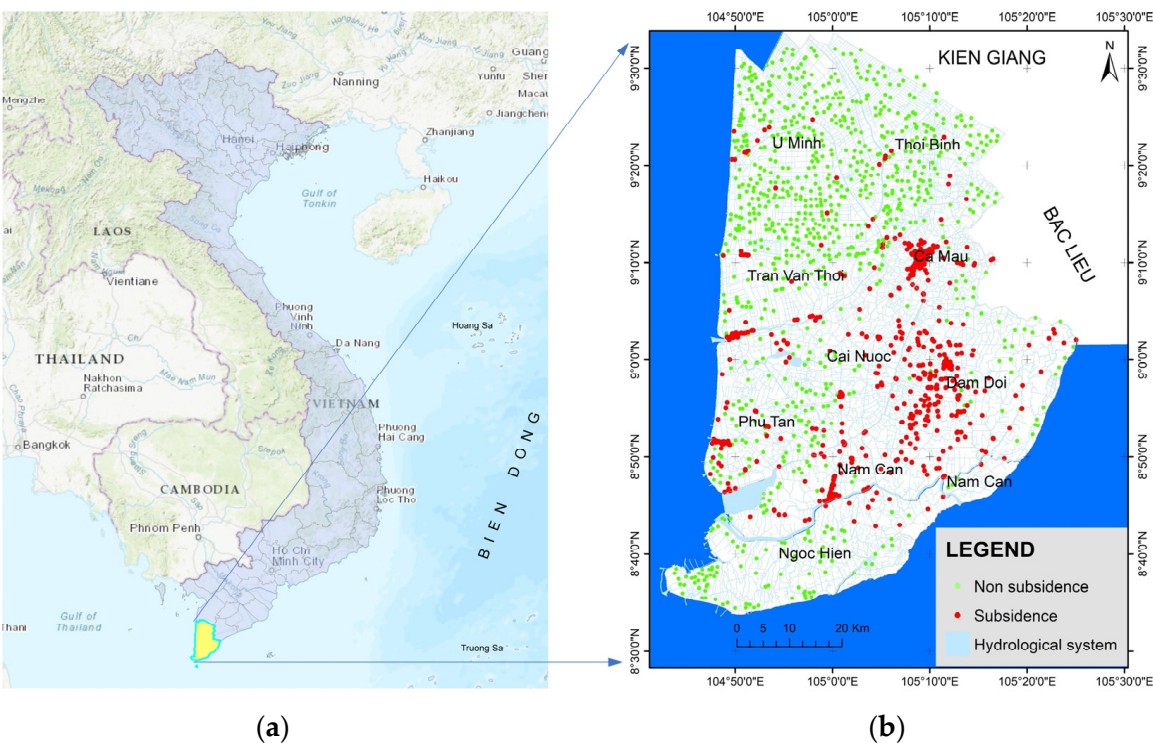

(**a**)　　　　　　　　　　　　　　　　(**b**)

**Figure 1.** Ca Mau research area on the map of Vietnam. (**a**) Vietnam map; (**b**) the boundary of Ca Mau Province is overlaid with the hydrosystem layer, along with sample points utilized in the subsidence susceptibility prediction model.

### 2.2. Hydrological Characteristics

The rivers, streams, and channels in Ca Mau Province form an intricate network, covering nearly 3% of the natural area. There are 8 main rivers and 3 primary canals with river mouths ranging in width from 45 m (Cai Tau River) to 1800 m (Cua Lon River) and depths varying from 3 m (Bai Hap estuary) to 19 m (Bo De estuary of Cua Lon River) [11].

### 3. Research Methodology

Boosting is a machine learning technique utilized to enhance the predictive ability of a machine learning algorithm by focusing on learning from more challenging cases. It operates by generating iterations of the original machine learning model and concentrating on addressing misclassified instances from the previous model until a desired level of accuracy is achieved [12]. There are several different types of boosting algorithms, but in this study, three boosting algorithms utilized with decision trees, namely AdaBoost, Gradient Boosting, and Extreme Gradient Boosting (XGBoost) were experimented with. Below, we sequentially introduce these three algorithms for the research area and select the optimal model.

### 3.1. AdaBoost

AdaBoost (Adaptive Boosting) is a popular machine learning ensemble technique used for classification and regression tasks and was invented by the authors [12]. It aims to improve the performance of weak learners (often referred to as "base classifiers" or "base models") by combining their predictions into a strong overall prediction. The core idea behind AdaBoost is to give more weight to instances that are misclassified by the previous base models, thereby focusing on the difficult cases.

Let us assume a binary classification problem with a target variable consisting of two labels: $y \in \{-1, 1\}$. Following the boosting method, the predictive function for an input

variable $x_i$ is denoted as $\hat{f}(x_i) \in \{-1, 1\}$, and the target variable y takes one of two values: $\{-1, 1\}$. In this case, the training error can be defined as follows [12]:

$$r = \frac{1}{N}\sum_{i=1}^{N} 1(y_i \neq \hat{f}(x_i))$$
(1)

where the following definitions hold:

$N$ is the number of training samples.

$\alpha_i$ represents the weight associated with the *i*-th training samples.

$y_i$ is the actual target value for the *i*-th samples.

According to Figure 2, the weak models are combined by assigning weights to each based on their performance. Stronger models are given higher weights in making the final predictions $\hat{f}(x_i)$ [13].

$$\hat{f}(x) = \text{sign}\left[\sum_{i=1}^{p} \alpha i \, \hat{f}^i(x)\right]$$
(2)

**Figure 2.** Diagram of the AdaBoost model. Each individual sub-model is trained on a dataset weighted according to calculations from the pre-model [13].

In the equation above, the sign(x) function is a function that takes the value 1 if the sign of x is positive and takes the value −1 otherwise.

### 3.2. Gradient Boosting

The Gradient Boosting (GB) algorithm was invented and introduced by Jerome H. Friedman in 2001, and it involves training weak models sequentially. However, instead of using the model error to weight the training data like AdaBoost, residuals are used [14].

Starting from the current model, GB tries to build a decision tree to match the residuals from the previous model. The special feature of this model is that instead of trying to match the target variable value of y, it will try to match the error value of the previous model. It then adds the training model to the prediction function to gradually update the residuals. Each decision tree in the model chain is very small in size with only a few decision nodes determined by the depth parameter d in the model. The Figure 3 illustrates this process in more detail.

### 3.3. XGBoost (Extreme Gradient Boosting)

XGBoost is an extremely powerful and popular machine learning model in both the machine learning and data science communities. XGBoost was mainly developed by Tianqi Chen and first announced in 2015 [15]. It falls under the category of Gradient Boosting algorithms, designed to optimize the performance of prediction models, especially in regression and classification tasks.

XGBoost uses Gradient Descent [16] to optimize the model by continuously improving the decision trees. XGBoost is efficiently implemented and supports parallel computation,

which speeds up training on multi-core computers. The steps in building the XGBoost model are as follows:

- XGBoost starts by constructing a weak decision tree, possibly a very small one.
- Computing the gradient of the loss function: After it has a weak tree, XGBoost calculates the gradient of the loss function (typically mean squared error in regression or log loss in classification) with respect to the data points. This gradient reflects the discrepancy between the current predictions and the actual values.
- Building the next tree to reduce the gradient: XGBoost proceeds to construct another decision tree with the aim of optimizing the reduction in the gradient (the difference between predictions and actual values). This yields a new model with improved predictive performance compared to the previous one.
- Combining the new tree with previous trees: XGBoost integrates this new tree into the overall model in addition to the previously built trees, creating a stronger model.
- Iterating the process: This process is repeated until a predefined number of trees (or tree layers) is reached or when the loss function no longer decreases significantly.

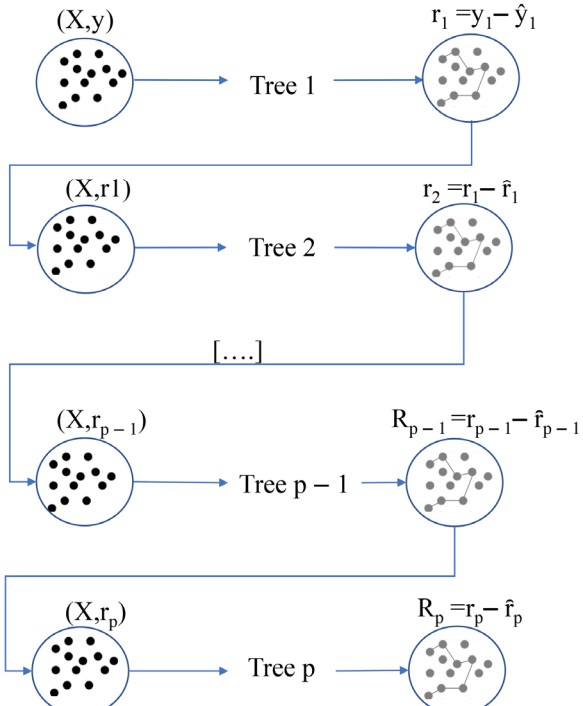

**Figure 3.** Training method using GB. Decision tree models are arranged in a sequence. Each decision tree is constructed based on the predictions of the preceding decision tree. At each decision tree, the model will attempt to fit the residuals from the previous decision tree [13].

The outstanding capabilities of XGBoost are as follows:

XGBoost has the advantages of avoiding overfitting by using techniques such as sub-sampling rows, columns, column per split levels, and applying regularized L1 and L2; resource utilization capability advantages including parallel computation on CPU/GPU, distributed computation across multiple servers, computation under resource constraints, and cache optimization to speed up training; and finally, the ability to handle missing data values and continue training using a previously built model to save time.

## 4. Data

### 4.1. The Inventory Points of Land Subsidence

A land subsidence inventory plays a crucial role in constructing land subsidence susceptibility models, providing essential information about the status and extent of land subsidence in specific areas. These data, along with several influencing factors, form the ba-

sis for training supervised subsidence susceptibility models. As previously mentioned, the Ca Mau Peninsula is a vast and relatively flat region, leading to infrequent data collection points and limited coverage of subsidence points. The total subsidence points collected by the Department of Survey, Mapping and Geographic Information Vietnam amount to 31 points [17]. However, the period of subsidence observations was intermittent, with the first measurement being in 2005 and the most recent being in 2020. Due to the scarcity of leveling measurement points, these points are only utilized as validation points for the model's prediction results. Additional land subsidence points were collected using multi-temporal radar image processing to obtain sample points for building the forecast model.

The measured ground deformations from stacks of archive Sentinel-1 SAR imagery acquired in the reference period November 2014–January 2019 were utilized to detect displacements caused by land subsidence and to estimate the subsidence's average velocity over the reference period from the document EMSN-062: Assessing changes in ground subsidence rates, Mekong Delta, Vietnam [18,19]. Land subsidence points made using the PSI method have been proven to have satisfactory accuracy [19]. The selection of prominent subsidence points using the PSI method relies on the criterion of points exhibiting a subsidence rate surpassing −1 cm/year. A total of 1011 subsidence points were identified based on this criterion. To be incorporated into the predictive model, non-subsidence points are also required apart from subsidence points. The 1000 non-subsidence points were selected from the PSI dataset including points with subsidence less than −1 cm/year and between −0.2 cm/year and +0.2 cm/year. The selection criterion for this range is due to the abundance of PS points in the dataset, while our allocation is limited to 1000 points, prompting the selection of points closest to the value of 0. Consequently, the model encompasses a total of 2011 points. The distribution of these points is shown in Figure 1.

*4.2. Influence Factors in the Subsidence Susceptibility Model*

When constructing a machine learning model for predicting land subsidence, several influential factors need to be considered to ensure the accuracy and effectiveness of the model. In various studies within this field, factors like elevation, slope, aspect, plan curvature, topographic wetness index (TWI), distance from streams, lithology, NDVI, land use/land cover (LULC), oxidation of organic matter, and groundwater extraction have been employed for land subsidence models [3,20,21]. However, considering the terrain, the Ca Mau Peninsula is a sedimentary landmass in the Mekong Delta characterized by very low terrain elevation differentials. Hence, our study meticulously deliberated and largely excluded terrain-related factors such as slope, aspect, TWI, and plan curvature. Eight influencing factors were selected: topographic elevation, soil type, geology, groundwater, LULC, NDVI, distance to roads, and distance to rivers. Figure 4 shows these input factors.

- Topography:

The topography has a significant impact on land subsidence and related phenomena. It can affect subsidence in various ways, such as influencing water flow and the accumulation of organic and mineral waste in the soil. Sloping terrain can lead to inclined subsidence, making the top layer of soil prone to sliding [7,22]. However, as mentioned in the research area section, Ca Mau has low topography with an average elevation of less than 1 m, so there are almost no slopes. Therefore, in this study, we only consider an elevation layer as a representative of the topography. The terrain elevation layer is taken from the digital terrain model (DTM), which is interpolated from elevation points derived from the 1:50,000 scale topographic map of 2014 provided by the Center of Survey and Mapping Data—Vietnam Department of Mapping and Geographic Information [17].

- Geology:

The geological structure can affect the strength of the soil and its load-bearing capacity. Soil with layered structures, cracks, or weaknesses may be more susceptible to subsidence. Thus, geological data are an important input layer influencing land subsidence. In the Mekong Delta, sediment cores and sequence stratigraphic studies indicate that the coastal

zone and the adjacent subaqueous delta on the shelf predominantly developed within the last ~1000 years. Consequently, the Mekong subaqueous delta is considered young compared to other Asian deltas [23].

According to research conducted by Ngoc et al. (2021) [24], the Mekong Delta provinces are characterized by Holocene sediments primarily composed of soft clay, ranging in thickness from a few meters to over 20 m, and occasionally exceeding 30–40 m. Ca Mau Province, a coastal lowland and a relatively young sedimentary area, exhibits thinner layers of soft clay compared to the interior provinces. The geological map with quaternary strata of Ca Mau is a 1:100,000 scale map provided by the Vietnam Institute of Geology and Mineral Resources and has been selected as an influence layer of the land subsidence susceptibility prediction model [25].

- Soil Type:

The type of soil can influence subsidence through its physical and chemical properties, including permeability, water retention, swelling and shrinking, hardness, and flexibility, as well as its interaction with groundwater. The soil's permeability affects the water infiltration rate through the soil. Soil with good permeability can lead to rapid water loss, contributing to the subsidence process. The properties of soil particles, such as clay, sand, and gravel, can affect changes in soil volume, thus affecting the likelihood of subsidence. Furthermore, factors such as soil layer stiffness, flexibility, and thickness also contribute to subsidence effects. The soil map of Ca Mau is a 1:100,000 scale map provided by the Vietnam Institute of Geology and Mineral Resources [25].

- LULC (Land Use and Land Cover):

This refers to how humans use the land, such as planting crops, building houses, constructing roads, urbanization, agricultural production, and afforestation. Land use can change over time due to human activities. Changes in surface cover can impact the water balance in the soil. Constructing urban areas, roads, or impermeable surfaces can cause changes in groundwater flow, affecting water balance and causing land subsidence. The LULC map of Ca Mau is a 1:50,000 scale map for 2020 provided by the Center of Survey and Mapping Data—Vietnam Department Of Survey, Mapping and Geographic Information [17].

- NDVI (Normalized Difference Vegetation Index):

NDVI is a commonly used index for measuring and analyzing the vegetation status on the ground based on satellite imagery. NDVI is widely used in areas such as land resource management, agriculture, environmental monitoring, and climate change observation. The NDVI is calculated from two wavelengths of light reflected from the ground, namely near-infrared (NIR) and red. The formula for calculating the NDVI is shown below [26].

$$NDVI = \frac{(NIR + RED)}{(NIR - RED)} \tag{3}$$

The NDVI (normalized difference vegetation index) typically ranges from −1 to +1. Negative values (often close to −1) usually appear over areas of water, rocks, snow, urban areas, or regions devoid of vegetation. Values close to 0 indicate areas with sparse or no vegetation, while positive values (often close to +1) signify the presence of abundant and well-developed vegetation.

The NDVI helps monitor changes in vegetation and soil conditions. When vegetation is dense, such as in dense forests or areas with full tree coverage, various interacting factors can contribute to stabilizing the soil and reducing subsidence. This is because plants with strong and dense root systems can create a useful network to firmly hold the soil. Roots help establish cohesion between soil particles, making the soil stronger and less susceptible to being eroded by water flow. The NDVI map of the Ca Mau area was generated using Sentinel-2 satellite images from July to August 2021 on the Google Earth Engine (GEE) platform and subsequently downloaded directly from the platform. GEE is developed by

Google to analyze and process satellite images and geographic data from various sources on Earth. This platform provides powerful tools for performing analysis of satellite images and geographic data from many different sources.

- Groundwater Depth:

Groundwater is a factor that can be considered one of the most crucial in influencing land subsidence. Numerous studies, such as those conducted by the authors of [10,27,28], have demonstrated the relationship between groundwater and land subsidence. Therefore, the groundwater depth dataset is a significant layer included in this research.

In Ca Mau Province, the primary source of water for both domestic and industrial purposes is groundwater. Groundwater resources meet the current water demands of the province, extracted from various types of wells and boreholes with different depths, diameters, and layers [29]. According to information from the research conducted by Thanh.,D.U and colleagues (2019) [30], it has been revealed that there are seven aquifers present within the Ca Mau Peninsula: the Holocene layer (qh), the Upper Pleistocene layer (qp$_3$), the Middle–Upper Pleistocene layer (qp$_{2-3}$), the Lower Pleistocene layer (qp$_1$), the Middle Pliocene layer (n$_2^2$), the Lower Pliocene layer (n$_2^1$), and the Upper Miocene layer (n$_1^3$). The potential groundwater exploitation area focuses on four main aquifers: qp$_{2-3}$, qp$_1$, n$_2^2$, and n$_2^1$. While the qp$_3$ and n$_1^3$ aquifers may contain good-quality groundwater, they are considered secondary for the following reasons: The qp$_3$ layer has a small thickness, only suitable for small-scale household exploitation [31]. Although the n$_1^3$ layer has a large thickness and is rich in water, its deep distribution leads to high exploitation costs, making it less common for utilization. In addition, in 2017, Minderhoud [32] demonstrated that the decline in groundwater levels led to very high land subsidence in the 25 years from 1991 to 2016 in the Mekong Delta region. In this article, the groundwater levels in different aquifers were interpolated. He proved that the qp$_{2-3}$ layer has the highest decline in groundwater level.

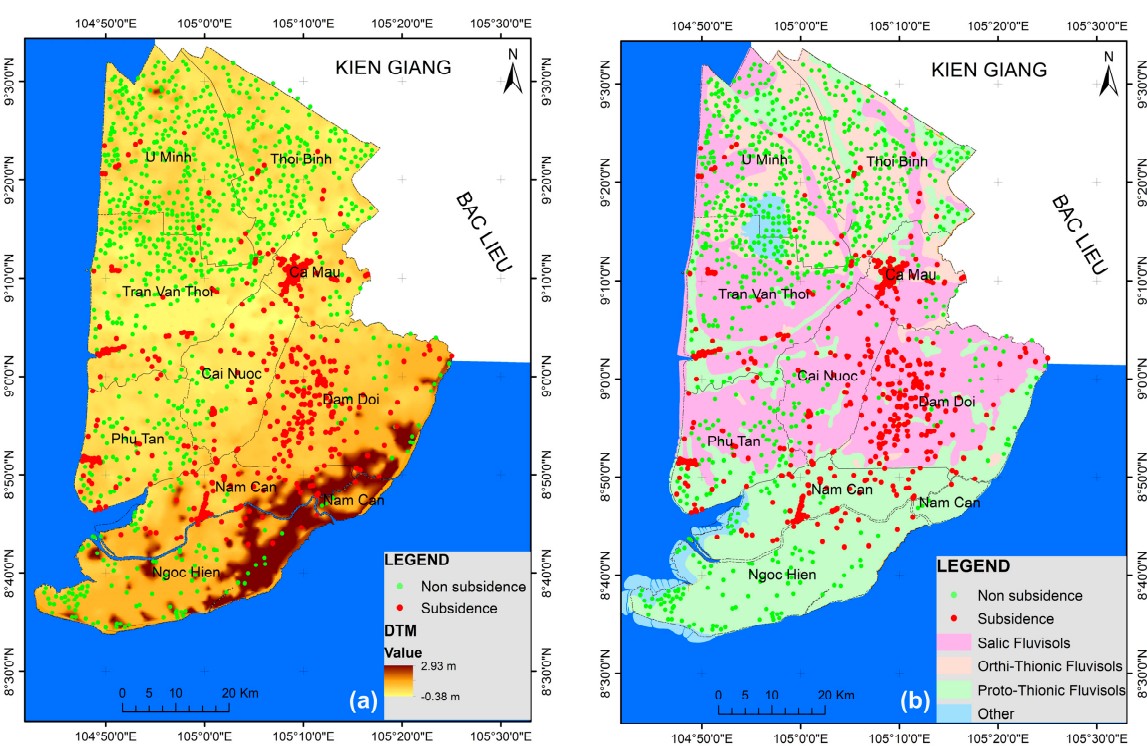

**Figure 4.** *Cont.*

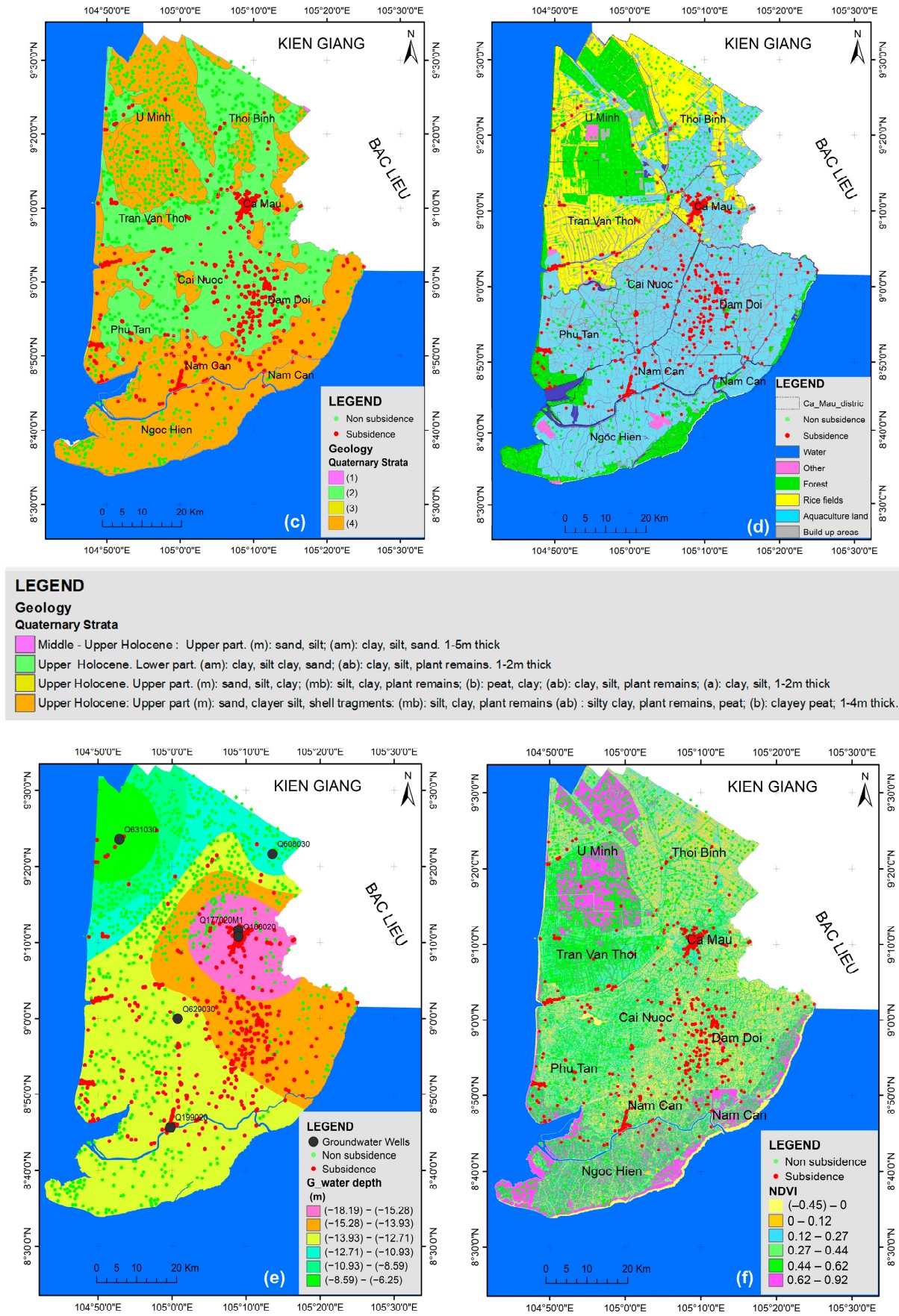

**Figure 4.** *Cont.*

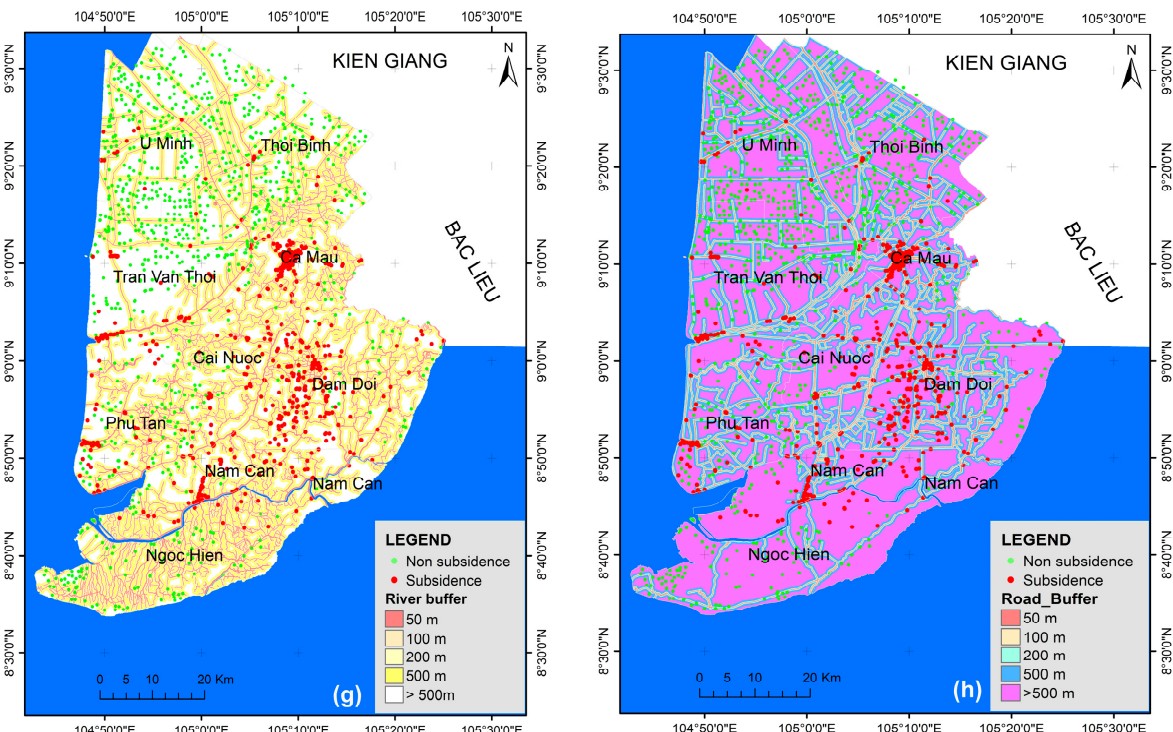

**Figure 4.** Input factor layers for the subsidence prediction model in Ca Mau area. (**a**) Elevation, (**b**) soil, (**c**) geology, (**d**) LULC, (**e**) groundwater depth, (**f**) NDVI, (**g**) distance to rivers, (**h**) distance to roads.

The data of groundwater extraction wells averaged monthly over 3 years, 2019, 2020, and 2021, in the $qp_{2-3}$ layer were used for the impact factor for the model. These data were provided by the National Center for Water Resources Planning and Investigation—Vietnam [29]. With these data, the groundwater depth map of the $qp_{2-3}$ layer was interpolated by the Kriging method [33] (Figure 4e).

- Distance to roads:

Subsidence often occurs near roads due to changes in the natural drainage system of the area during road construction. The construction of drainage ditches or alterations in the landscape can reduce the natural drainage ability of the environment, leading to flooding and an increased risk of subsidence. Additionally, traffic activities on the road can exert additional load on the ground. Vehicles moving on the road generate impacts and pressure on the soil surface, compressing the soil more and contributing to subsidence. The road map is taken from the 1:50,000 scale topographic map for 2010 provided by the Center of Survey and Mapping Data—Vietnam Department Of Survey, Mapping and Geographic Information [17]. This map is a bit old; therefore, we have gathered data from the OpenStreetMap (OSM) source [34]. The tertiary roads have been incorporated to enhance the map originally provided by the Department of Mapping and Geographic Information. Roads were buffered according to distance at different levels: 50 m, 100 m, 200 m, and 500 m.

- Distance to rivers:

The presence of water bodies can increase moisture in the surrounding environment. Moist soil is more susceptible to compression and may lead to subsidence. Human activities creating infrastructure around water bodies, such as building drainage systems, bridges, or urban areas, can also affect soil characteristics and contribute to the subsidence process.

Ca Mau possesses an elaborate network of rivers and canals, establishing its prominence as the leading province in the Mekong Delta region with a total length of large and

small canals reaching up to 7000 km. The river map is also derived from the 1:50,000 scale topographic map issued in 2010 by the Center of Survey and Mapping Data—Vietnam Department of Survey, Mapping, and Geographic Information [17]. Rivers were buffered at various distances, including 50 m, 100 m, 200 m, and 500 m.

*4.3. Data Standardization*

When constructing a model for predicting land subsidence, it is essential to standardize the input data by converting them to a consistent parameter system (referred to as data standardization). Variations in measurement units or a wide range of values in the data can potentially impact the model. Data standardization mitigates this effect by bringing all features to a uniform scale. In this study, the data normalization is achieved through the utilization of the Frequency Ratio (FR) method [35] that relies on the spatial correlation between previous instances of land subsidence and the factors contributing to the occurrence of land subsidence. A higher FR value signifies a more robust correlation between subsidence occurrences and the influencing factors. The FR value is computed using the formula (4) [36]. Using ArcGIS 10.8 software, the data for the 1011 selected subsidence points mentioned above were calculated and standardized according to Table 1.

$$FR = \frac{Npix(1)/Npix(2)}{\sum Npix(3)/\sum Npix(4)} \tag{4}$$

where *Npix* (1) is the number of land subsidence pixels of the factor class, *Npix* (2) is the total number of pixels of the sub-class over the entire study area, *Npix* (3) is the total number of land subsidence pixels of the study area, and *Npix* (4) is the total number of pixels of the study area.

**Table 1.** Land subsidence conditioning factors and their classification.

| Factor | Sub-Class | LS Points (Npix(1)) | %Land Subsidence | Class Pixels (Npix(2)) | % Class Pixels | FR |
|---|---|---|---|---|---|---|
| Elevation (m) (natural breaks) | (1) [−0.38–0.33] | 139 | 13.75 | 38,040 | 36.909 | 0.373 |
| | (2) [0.33–0.68] | 481 | 47.58 | 32,500 | 31.534 | 1.509 |
| | (3) [0.68–1.11] | 314 | 31.06 | 20,603 | 19.990 | 1.554 |
| | (4) [1.11–1.61] | 50 | 4.95 | 4999 | 4.850 | 1.020 |
| | (5) [1.61–2.11] | 27 | 2.67 | 6646 | 6.448 | 0.414 |
| | (6) [2.11–2.93] | 0 | 0.00 | 276 | 0.268 | 0.000 |
| Soil | (1) Proto-Thionic Fluvisols | 499 | 49.36 | 36,193 | 35.117 | 1.406 |
| | (2) Orthi-Thionic Fluvisols | 30 | 2.97 | 18,965 | 18.401 | 0.161 |
| | (3) Salic Fluvisols | 482 | 47.68 | 43,683 | 42.384 | 1.125 |
| | (4) Other | 0 | 0.00 | 4223 | 4.097 | 0.000 |
| Geology | (1) | 3 | 0.30 | 28 | 0.027 | 10.922 |
| | (2) | 599 | 59.25 | 45,175 | 43.832 | 1.352 |
| | (3) | 4 | 0.40 | 549 | 0.533 | 0.743 |
| | (4) | 405 | 40.06 | 57,310 | 55.606 | 0.720 |
| Groundwater (natural breaks) | (1) [(−18.19 m)–(−15.29 m)] | 254 | 25.12 | 10,133 | 9.832 | 2.555 |
| | (2) [(−15.28 m)–(−13.93 m)] | 261 | 25.82 | 21,721 | 21.075 | 1.225 |
| | (3) [(−13.93 m)–(−12.71 m)] | 459 | 45.40 | 48,261 | 46.826 | 0.970 |
| | (4) [(−12.71 m)–(−10.93 m)] | 16 | 1.58 | 10,458 | 10.147 | 0.156 |
| | (5) [(−10.93 m)–(−8.59 m)] | 2 | 0.20 | 6213 | 6.028 | 0.033 |
| | (6) [(−8.59 m)–(−6.25 m)] | 19 | 1.88 | 6278 | 6.091 | 0.309 |
| NDVI (natural breaks) | (1) [(−0.45)–(−0.0)] | 11 | 1.09 | 7851 | 7.618 | 0.143 |
| | (2) [(−0.0)–0.12] | 450 | 44.51 | 18,127 | 17.588 | 2.531 |
| | (3) [0.12–0.27] | 279 | 27.60 | 21,300 | 20.667 | 1.335 |
| | (4) [0.27–0.44] | 196 | 19.39 | 23,059 | 22.373 | 0.867 |
| | (5) [0.44–0.62] | 66 | 6.53 | 16,305 | 15.820 | 0.413 |
| | (6) [0.62–0.92] | 9 | 0.89 | 16,422 | 15.934 | 0.056 |

**Table 1.** *Cont.*

| Factor | Sub-Class | LS Points (Npix(1)) | %Land Subsidence | Class Pixels (Npix(2)) | % Class Pixels | FR |
|---|---|---|---|---|---|---|
| LULC | (1) Water | 0 | 0.00 | 42,851 | 41.577 | 0.000 |
| | (2) Alluvial land | 3 | 0.30 | 570 | 0.553 | 0.537 |
| | (3) Forest | 18 | 1.78 | 7787 | 7.555 | 0.236 |
| | (4) Rice fields | 324 | 32.05 | 17,234 | 16.722 | 1.917 |
| | (5) Aquaculture land | 115 | 11.37 | 31,927 | 30.978 | 0.367 |
| | (6) Built-up areas | 551 | 54.50 | 2695 | 2.615 | 20.842 |
| Distance to Road | (1) [0–50 m] | 209 | 20.67 | 4056 | 3.935 | 5.253 |
| | (2) [50–100 m] | 185 | 18.30 | 3569 | 3.463 | 5.284 |
| | (3) [100–200 m] | 272 | 26.90 | 6361 | 6.172 | 4.359 |
| | (4) [200–500 m] | 181 | 17.90 | 15,495 | 15.034 | 1.191 |
| | (5) [>500 m] | 164 | 16.22 | 73,583 | 71.395 | 0.227 |
| Distance to River | (1) [0–50 m] | 190 | 18.79 | 9576 | 9.291 | 2.023 |
| | (2) [50–100 m] | 115 | 11.37 | 7425 | 7.204 | 1.579 |
| | (3) [100–200 m] | 181 | 17.90 | 13,456 | 13.056 | 1.371 |
| | (4) [200–500 m] | 219 | 21.66 | 28,456 | 27.610 | 0.785 |
| | (5) [>500 m] | 306 | 30.27 | 44,151 | 42.838 | 0.707 |

After the data had been normalized, the factor maps were incorporated into the model, comprising eight layers. The FR data column in Table 1 was utilized to train the model.

## 5. Results and Discussion

### 5.1. Construction of the Models

The subsidence data consist of 2011 PSI points incorporated into the model. Additionally, eight influential data layers, namely elevation, soil type, geology, groundwater, land use and land cover (LULC), normalized difference vegetation index (NDVI), distance to roads, and distance to rivers, are included. When employing machine learning models, the dataset is split into two distinct parts: the training and the testing. This division is crucial for evaluating the model's performance on new data not previously encountered during training. The training set constitutes 70% of the input data, while the testing set comprises the remaining 30%. Within the dataset, values are labeled as 1 (indicating the presence of subsidence) or 0 (indicating non-subsidence). Figure 5 depicts the process of constructing a prediction model utilizing three boosting methods: AdaBoost, Gradient Boosting, and XGBoost. Python served as the programming language used for this research.

For machine learning models, setting hyperparameters is an essential step in constructing and refining the model. Hyperparameters play a direct role in determining model performance. By fine-tuning their values, we can enhance the model's predictive capacity and reduce errors. Adjusting hyperparameters such as tree depth, learning rate, and regularization parameters enables us to mitigate overfitting and enhance the model's ability to generalize. Models have numerous hyperparameters, yet not all parameters require adjustment. Below is an informational table showcasing the parameters after testing and adjustments to optimize the models in the study area (Table 2).

**Table 2.** Hyperparameters optimized for the models in the study area.

| | N_Estimators | Learning_Rate | Max_Depth | Loss | Lambda | Alpha |
|---|---|---|---|---|---|---|
| XGBoost (XGB) | 600 | 0.003 | 5 | Squared error | 3 | 0.01 |
| Gradient Boosting (GB) | 600 | 0.003 | 5 | Squared error | | |
| AdaBoost model (ADB) | 600 | 1 | | Squared error | | |

In ensemble algorithms, n_estimators determine the number of base learners created and combined to form a strong learner. Each base learner is typically a decision tree in

AdaBoost and Gradient Boosting or an improved decision tree in XGBoost. The number of n_estimators must be controlled within a reasonable range to avoid overfitting. In our case study, the number of n_estimators was taken to be 600 after several trials.

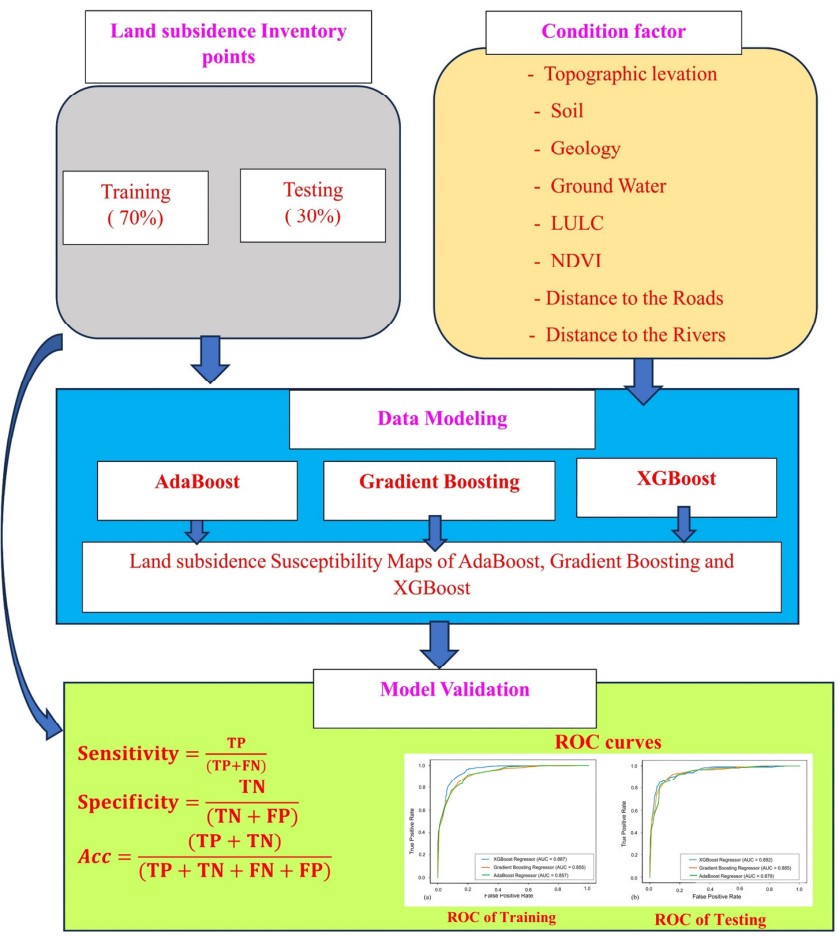

**Figure 5.** Processing flow chart for land subsidence susceptibility mapping [7].

The max_depth (maximum depth of the tree) in Gradient Boosting and XGBoost depends on several factors. While increasing the depth of the tree can help the model learn more complex relationships between input variables and the target, it also increases the risk of overfitting. In the study, we experimented and selected 5 as the max_depth for both the GB and XGB models. For AdaBoost, this parameter does not exist.

Learning_rate is an important hyperparameter that adjusts the learning rate of the model. It determines the magnitude of the model's step when updating its weights during training. It is usually set to a value less than or equal to 1. After experimenting with learning_rate values, 0.03 was selected.

In the XGBoost model, alpha and lambda are two hyperparameters used to control the regularization process and reduce overfitting. Both of these hyperparameters apply regularization to the model by adding penalty terms to the loss function. Alpha is the coefficient applied to the L1 regularization component in the XGBoost loss function. Lambda is the coefficient applied to the L2 regularization component in the XGBoost loss function. The values of Lambda and Alpha are chosen to be 3 and 0.01, respectively. The mean squared error function is selected as the loss function for all three models.

### 5.2. Evaluation of the Importance of the Model's Input Variables

Determining the importance of variables helps gain insights into the contribution of each variable to the predicted outcome. This provides an overall understanding of the influence of these factors. In addition, assessing variable importance aids in model

optimization. If certain variables have minimal impact on the prediction, consideration can be given to removing them, simplifying the model while retaining accuracy. Additionally, this work avoiding excessive use of correlated variables helps mitigate collinearity issues, where multiple explanatory variables are highly correlated.

Figure 6 presents a plot that summarizes the importance values of the input variables, elucidating their relationships with the predicted outcomes. The vertical axis in the chart represents the intensity of the impact of each input factor, while the horizontal axis denotes the respective factors. Higher values on the vertical axis indicate higher efficiency. From Figure 6, it can be understood that LULC and groundwater depth exhibit a significant level of influence on the prediction results compared to other results, and the XGBoost model (blue column) shows a quite high influence while the two models GB (orange column) and AdaBoost (grey column) show lower values. The next influencing factor is the distance to roads and soil. The reason can be understood as vehicle traffic and loads from road traffic can create pressure on the soil layer, especially when the soil is already weakened due to other reasons.

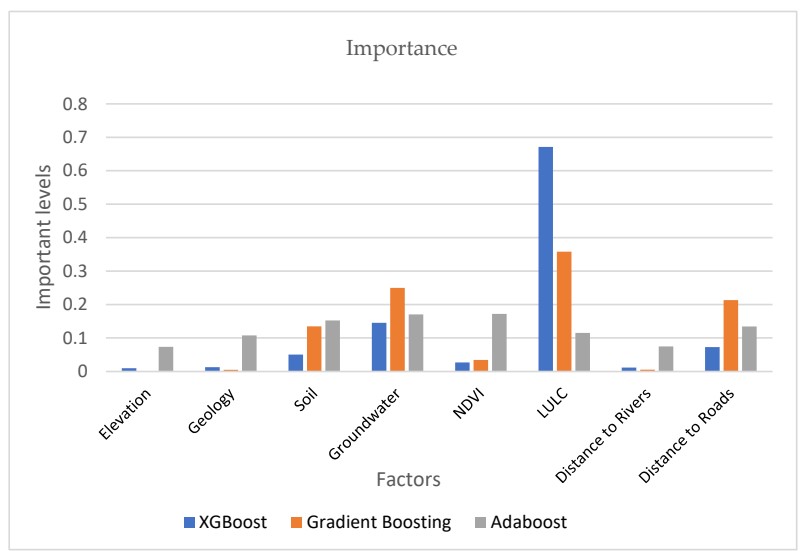

**Figure 6.** Evaluation of the importance of 8 input variables in three models.

Other factors that affect the model, although not much, cannot be ignored; for example, geology, distance to rivers and streams, and elevation have very little influence, and this is easy to explain because Ca Mau has a quite low topography, with many places having elevation lower than sea level.

*5.3. Evaluation of Model Performance*

To evaluate model performance, we use the receiver operating characteristic (ROC) curve, area under the curve (AUC), sensitivity, specificity, and accuracy (Acc).

The ROC curve is a graph that illustrates the relationship between the true positive rate (TPR) and false positive rate (FPR) of a classification model at different decision thresholds. TPR is the ratio of correctly predicted positive cases (true positives) to the total number of actual positive cases. FPR is the ratio of incorrectly predicted positive cases to the total number of actual negative cases.

AUC is the area under the ROC curve. AUC measures the ability of a classification model to correctly classify positive versus negative instances. AUC typically ranges from 0 to 1, with a higher AUC indicating a better model performance. The relationship between model performance and AUC can be quantified as follows: excellent (0.9–1), very good (0.8–0.9), good (0.7–0.8), fair (0.6–0.7), and poor (0.5–0.6) [37].

The accuracy assessment method using the ROC curve and AUC is a valuable tool for validating land subsidence prediction models. The utilization of the ROC curve and AUC

aids in evaluating result reliability, comparing the performances of different models, and identifying the best model for land subsidence prediction purposes.

In evaluating the performance of a predictive model, combining the ROC curve and AUC with other metrics such as accuracy, sensitivity, and specificity will provide a more comprehensive overview of the model's performance [38]. The formula for calculating sensitivity is as follows:

$$\text{Sensitivity} = \frac{\text{TP}}{(\text{TP} + \text{FN})} \qquad (5)$$

where TP (true positive) is the number of true positive instances correctly identified and FN (false negative) is the number of false negative instances incorrectly identified.

The formula for calculating specificity is as follows:

$$\text{Specificity} = \frac{\text{TN}}{(\text{TN} + \text{FP})} \qquad (6)$$

TN (true negative) is the number of true negative instances correctly identified, and FP (false positive) is the number of false positive instances incorrectly identified.

Accuracy is calculated according to the following formula:

$$Acc = \frac{(\text{TP} + \text{TN})}{(\text{TP} + \text{TN} + \text{FN} + \text{FP})} \qquad (7)$$

For the Ca Mau research area, with three selected models ADB, GB, and XGB, the values are calculated based on the training set in Table 3 and the testing set in Table 4 below, along with the ROC curves for the training and testing datasets (Figure 7).

**Table 3.** Performance evaluation table of models on the training dataset.

|  | TP | TN | FP | FN | Sensitivity | Specificity | AUC | ACC |
|---|---|---|---|---|---|---|---|---|
| XGBoost (XGB) | 571 | 676 | 135 | 25 | 0.958 | 0.834 | 0.887 | 0.886 |
| Gradient Boosting (GB) | 581 | 622 | 125 | 79 | 0.88 | 0.833 | 0.855 | 0.855 |
| AdaBoost model (ADB) | 568 | 638 | 138 | 63 | 0.9 | 0.822 | 0.857 | 0.857 |

**Table 4.** Performance evaluation table of models on the testing dataset.

|  | TP | TN | FP | FN | Sensitivity | Specificity | AUC | ACC |
|---|---|---|---|---|---|---|---|---|
| XGBoost (XGB) | 275 | 263 | 19 | 47 | 0.854 | 0.933 | 0.892 | 0.891 |
| Gradient Boosting (GB) | 248 | 287 | 46 | 23 | 0.915 | 0.862 | 0.885 | 0.886 |
| AdaBoost model (ADB) | 271 | 259 | 23 | 51 | 0.842 | 0.918 | 0.879 | 0.877 |

With three prediction models, the accuracy of all three models is very high when the area under the curve (AUC) is greater than 0.8. Among them, XGB achieves the highest accuracy when AUC > 0.88 for both the training and testing datasets. Sensitivity measures the model's ability to accurately identify cases of subsidence within the total number of actual subsidence cases. Therefore, high sensitivity also implies accurately predicting subsidence points distributed in the research area. According to Tables 3 and 4, sensitivity is greater than 0.8, indicating that the accuracy of predicting subsidence points is very high for the three models.

Specificity is a metric that measures the ability of a model to accurately identify non-subsidence cases within the total number of actual non-subsidence cases in the research area. Also, according to Tables 3 and 4, we can easily observe that on the training set, the accuracy of predicting non-subsidence points is consistently greater than 0.8. However, when considering three prediction models on the testing dataset, the GB model has the highest sensitivity but the lowest specificity, while the ADB model has the highest specificity but the lowest sensitivity. This indicates an imbalance in predicting points of subsidence

susceptibility and non-subsidence susceptibility in these two models, especially on the testing dataset because this is the one that did not participate in the training model.

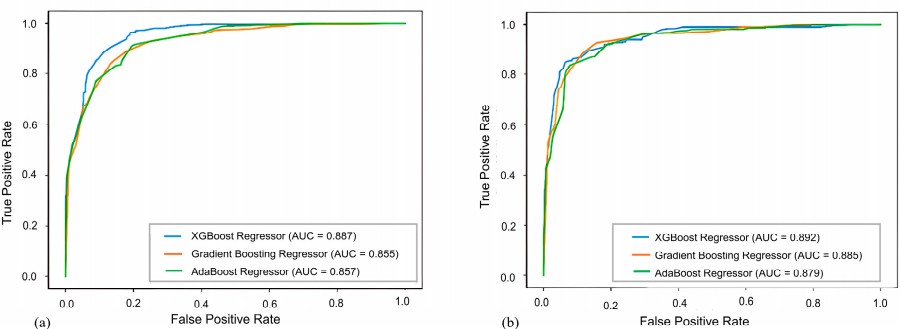

**Figure 7.** ROC curves and AUC values of models: (**a**) training dataset and (**b**) testing dataset.

All three models exhibit high overall accuracy (Acc) for the training and testing datasets, exceeding 0.8, with XGB having slightly higher accuracy. Examining the values of area under the curve (AUC), XGB achieves the highest values for both the training and testing datasets. This may indicate that the performance of the XGB model is superior and could be chosen for creating maps predicting land subsidence susceptibility.

The XGB model was utilized for predicting a land subsidence susceptibility map, and the result was exported as TIFF file and further refined using ArcGIS 10.8 software. Employing a classification method to categorize data classes manually in ArcGIS 10.8, the subsidence susceptibility map was divided into five levels, namely "Very Low", "Low", "Moderate", "High", and "Very High", corresponding to values "<0.3", "0.3–0.5", "0.5–0.7", "0.7–0.9", and "0.9–1", respectively [39]. The distribution of subsidence susceptibilities from the XGB model is illustrated in Figure 8. In this figure, the map illustrates the subsidence susceptibility across the province (Figure 8a), and a closer view of an area exhibiting significant subsidence around Ca Mau City and some adjacent districts to the city's south, as shown in Figure 8b. Additionally, we calculate the percentage of subsidence susceptibility based on the categorized levels as depicted in Figure 8c.

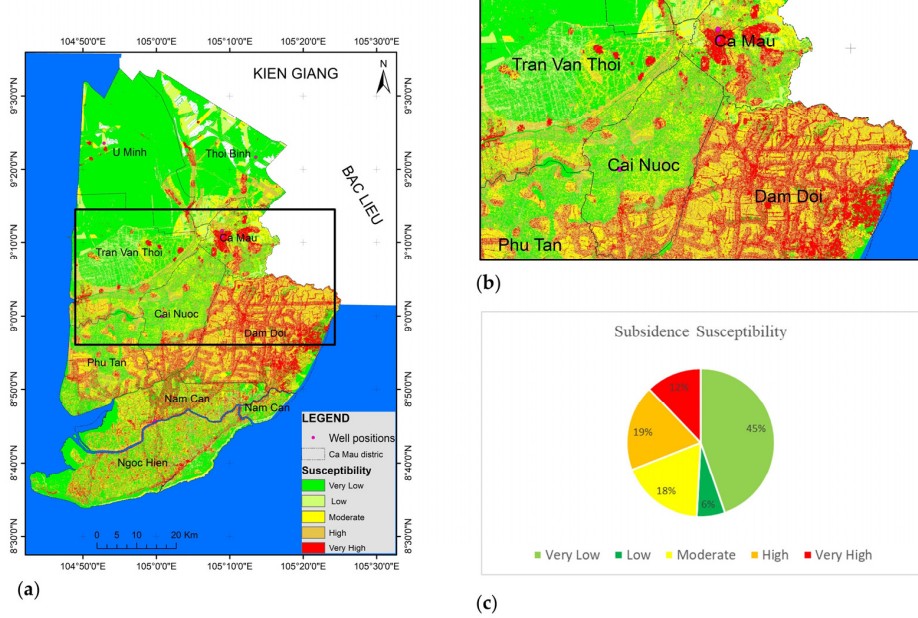

**Figure 8.** (**a**) Land subsidence susceptibility distribution map using the XGB model; (**b**) land subsidence susceptibility distribution inside the black rectangle in (**a**); (**c**) distribution of land subsidence susceptibility areas by percentage.

### 5.4. Discussion

For the selected XGB model, the very high land subsidence susceptibility concentration covers 12% of the total provincial area. It is distributed around Ca Mau City and the surrounding areas to the south of Ca Mau City. This is also explained by the significant influence of land use and land cover (LULC) on the predictive model (Figure 6). The FR values for the sub-classes of LULC in Table 1 reveal a clear pattern. Specifically, the built-up area class exhibits the highest FR value, indicating a strong association between land subsidence and this land type. The subsidence susceptibility in built-up areas is ten-fold greater than that in rice fields and hundreds of times larger than that in forested land. The conversion of land use from ponds or rice fields to urban areas has resulted in numerous problems, including decreased groundwater levels. This, coupled with the heavy load of buildings and infrastructure, has led to significant subsidence compared to other regions [40].

High land subsidence susceptibility is concentrated in the southeast of Ca Mau City, adjacent to Bac Lieu Province. In this area, the land is relatively low and primarily used for aquaculture. High subsidence points are also concentrated along transportation routes, which is more easily explained as training points are often measured along these routes. On the other hand, the subsidence sample points, derived from satellite images using the PSI method, as mentioned in the data collection section, are mainly acquired with high-persistence scattering and are more clustered around artificial structures such as roads, bridges, and buildings rather than other locations.

The moderate land subsidence susceptibility is 18% of the scattered distribution, not concentrated. Meanwhile, low land subsidence susceptibility covers 6%, mainly located in the western part of the Ca Mau Peninsula in Tran Van Thoi district. The very low land subsidence susceptibility is concentrated in the north of Ca Mau and covers the largest area at 45%. This northern area includes the U Minh district and part of the Tran Van Thoi district, where the majority of the U Minh Ha forest is located. The U Minh Ha forest is a unique ecosystem with alternating saline and freshwater, creating a distinctive environment for various plant species. In this area, only a few points have high land subsidence susceptibility, mainly near the groundwater extraction wells.

The soil types have little influence on the land subsidence susceptibility because the predominant soil types in the Ca Mau region are Proto-Thionic Fluvisols and Salic Fluvisol. The Ca Mau City area is a typical example of Proto-Thionic fluvial soil, while the soil in the southern part of Ca Mau City at Cai Nuoc, Dam Doi, and Tran Van Thoi is Salic Fluvisol. Both of these soil types exhibit a high to very high susceptibility to land subsidence in the research area.

Furthermore, as previously mentioned, the distribution of subsidence susceptibility is mainly concentrated along traffic routes, particularly in the area stretching from Ca Mau City to the south. While the river system in this region is notably dense, its influence on subsidence susceptibility appears to be negligible. This explains the higher impact level of roads in comparison to rivers (see Figure 6).

Related to the depth of the groundwater, the profile (A–B line) (Figure 9b) is taken (Figure 9a), revealing that the groundwater depth has significantly decreased in the Ca Mau City area compared to the U Minh area. The decline in groundwater levels is a major contributing factor to land subsidence. Therefore, the possibility of land subsidence in the Ca Mau City area is very high, and this has been shown very clearly on the subsidence susceptibility map made from XGB.

In region A, the land subsidence susceptibility is generally low, indicated by the green color mainly in this area, except for localized areas with higher subsidence potential, notably concentrated around roads and near the groundwater well fields in the U Minh area, denoted by the magenta dot symbol. In contrast, the situation in region B, encompassing Ca Mau City, differs significantly. Here, the susceptibility of high and very high subsidence is more pronounced, particularly in proximity to two groundwater exploitation wells. This observation underscores the strong correlation between groundwater depth presence,

traffic patterns, and land subsidence susceptibility, which aligns with the assessment of the influencing factors' importance level that is mentioned in Figure 6. Furthermore, while analyzing the A–B profile, it becomes apparent that the depth of the groundwater level differs between the two locations. At point A, situated in the U Minh area, the depth is approximately −6 m, whereas at point B, located in the center of Ca Mau City, it is approximately −16 m. This discrepancy underscores the influence of land use and groundwater exploitation in these respective areas. In region A, characterized primarily by forest cover, and region B, mostly urban, the demand for groundwater for daily human consumption and industrial and agricultural activities is notably higher.

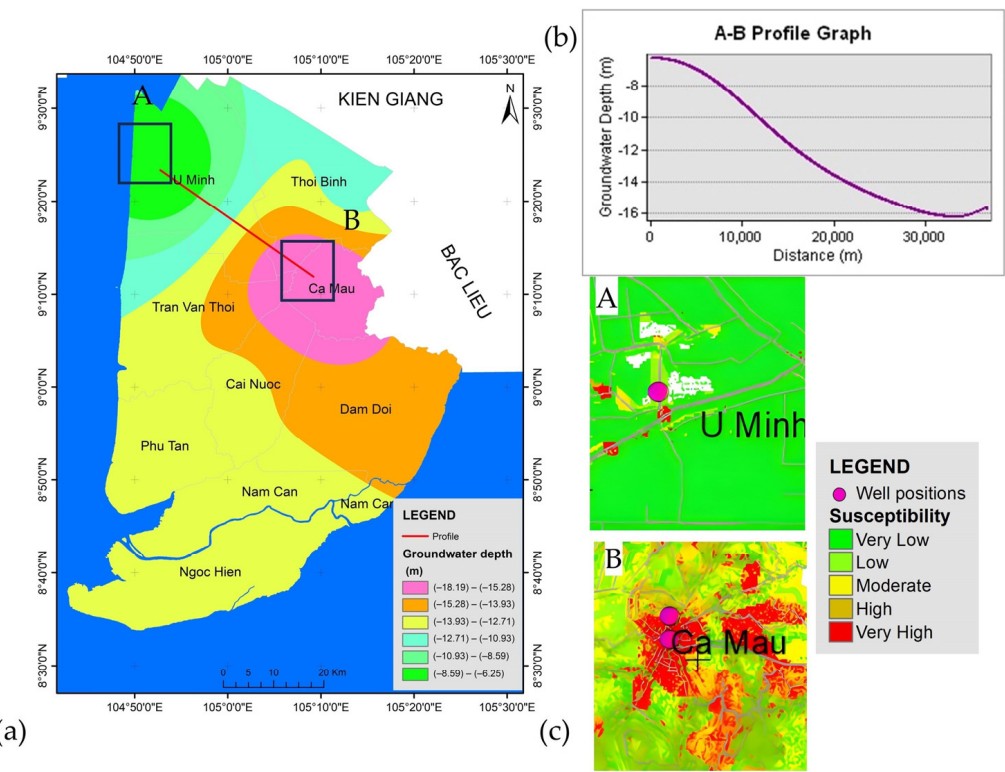

**Figure 9.** (**a**) Map of average groundwater levels over the years for aquifer qp$_{2-3}$; (**b**) a cross-section depicting the depth of the groundwater; (**c**) location of the land subsidence susceptibility map zoomed in at positions A and B with well positions.

Consequently, groundwater depletion is more severe in Ca Mau City's urban area than in the forested region of U Minh, further affirming the primary factors contributing to subsidence susceptibility and justifying our accurate selection of impact factors for the model.

To further assess the validity of the land subsidence susceptibility predictions generated by the XGB model, 31 subsidence points measured using second- and third-order leveling techniques were utilized. These points were not utilized during the model-building process; instead, they were solely employed to evaluate the model's accuracy. Elevations of these points were measured in two periods: 2005 and 2020. The subsidence rate, expressed as the change in elevation over 15 years, is presented in the "Subsidence rate/year" column in Table 5. The greatest subsidence values are predominantly concentrated in the center of Ca Mau City, extending along the roads. These subsidence values were then classified into different levels for comparison with the subsidence susceptibility map generated by the XGB model: "Very Low", "Low", "Medium", "High", and "Very High". Average annual subsidence values exceeding −1.5 cm/year were categorized as "Very High", while values ranging from −1 cm to −1.5 cm were classified as "High", and so forth. These measurement points were then overlaid onto the subsidence susceptibility prediction map generated

by XGB. The blue color on the map indicates that leveling points with high subsidence values are concentrated in Ca Mau City and districts such as Dam Doi and Nam Can. Some points with high subsidence (cyan) are situated along traffic routes, with many distributed in the southern region of Ca Mau City. Additionally, several points in the northern part of Ca Mau City are depicted in Figure 10b. In this area, there are points III(CH-HP)1 and II(SC-PL)18 with level measurement values showing high subsidence, but in the map, the subsidence susceptibility is low and very low. This shows that errors in predicted results using XGB have appeared in this area.

**Table 5.** Comparison between leveling survey and subsidence susceptibility [17].

| ID | Benchmark Name | Latitude (Degree) | Longitude (Degree) | Subsidence Rate/Year 2005–2020 (cm) | Class of Subsidence Rate | Susceptibility | Compare |
|---|---|---|---|---|---|---|---|
| 1 | II(CM-TVT)4 | 9.12 | 105.03 | −1.90 | Very High | Very High | |
| 2 | II(CM-TVT)5 | 9.10 | 105.00 | −1.11 | High | High | y |
| 3 | II(CM-TVT)7A | 9.08 | 104.97 | −1.39 | High | High | y |
| 4 | II(CM-TVT)2 | 9.17 | 105.08 | −1.09 | High | Low | |
| 5 | II(NC-ĐH)22 | 9.00 | 105.08 | −0.78 | Moderate | Low | |
| 6 | II(NC-ĐH)23 | 9.03 | 105.21 | −1.71 | Very High | Very High | y |
| 7 | II(NC-ĐH)24 | 9.05 | 105.24 | −2.89 | Very High | Very High | y |
| 8 | II(NC-ĐH)25 | 9.08 | 105.24 | −1.82 | Very High | Very High | y |
| 9 | II(NC-ĐH)26 | 9.09 | 105.27 | −0.76 | Moderate | Low | |
| 10 | II(SC-PL)18 | 9.41 | 105.15 | −1.05 | High | Low | |
| 11 | II(SC-PL)24 | 9.18 | 105.15 | −2.52 | Very High | Very High | y |
| 12 | II(SC-PL)25A | 9.18 | 105.16 | −2.33 | Very High | Very High | y |
| 13 | II(SC-PL)27 | 9.16 | 105.22 | −2.35 | Very High | Very High | y |
| 14 | II(SC-PL)28 | 9.16 | 105.24 | −1.47 | High | High | y |
| 15 | II(TB-HĐB)1 | 9.42 | 105.12 | −0.59 | Low | High | |
| 16 | II(TVT- NC)1 | 9.06 | 105.00 | −0.59 | Low | Low | y |
| 17 | II(TVT- NC)2 | 9.04 | 105.03 | −1.41 | High | High | y |
| 18 | II(TVT- NC)4 | 8.97 | 105.01 | −0.77 | Moderate | Moderate | y |
| 19 | II(TVT- NC)5 | 8.94 | 105.01 | −1.69 | Very High | Very High | y |
| 20 | III(CH-HP)1 | 9.35 | 105.23 | −1.50 | Very High | Very Low | |
| 21 | III(CM-PD)2 | 9.10 | 105.14 | −2.49 | Very High | High | |
| 22 | III(HS-CM)4 | 9.31 | 105.09 | −1.19 | High | High | y |
| 23 | III(HS-CM)5 | 9.28 | 105.09 | −0.75 | High | High | y |
| 24 | III(HS-CM)6 | 9.26 | 105.08 | −0.79 | Moderate | Low | |
| 25 | III(HS-CM)7 | 9.23 | 105.08 | −1.43 | High | High | y |
| 26 | III(HS-CM)8 | 9.20 | 105.09 | −1.02 | High | High | y |
| 27 | III(HS-CM)9 | 9.20 | 105.12 | −1.41 | High | High | y |
| 28 | III(PĐ-CN)3 | 8.98 | 105.15 | −1.60 | Very High | Very High | y |
| 29 | III(PĐ-CN)4 | 8.96 | 105.10 | −1.57 | Very High | Very High | y |
| 30 | III(PĐ-CN)6 | 8.93 | 105.05 | −1.27 | High | High | y |
| 31 | III(TV-PĐ)5 | 9.10 | 105.18 | −1.03 | High | High | y |

Upon comparing the model values with those obtained through leveling measurements (Table 5), it was found that out of the 31 points measured, 22 exhibited accurate predictions when compared with the XGB model (by marking "y" in the comparison column). This demonstrates a predictive accuracy of 70.9% in assessing subsidence susceptibility compared to standard measurement points. Thus, it is evident that the XGB model has successfully achieved its predicted results.

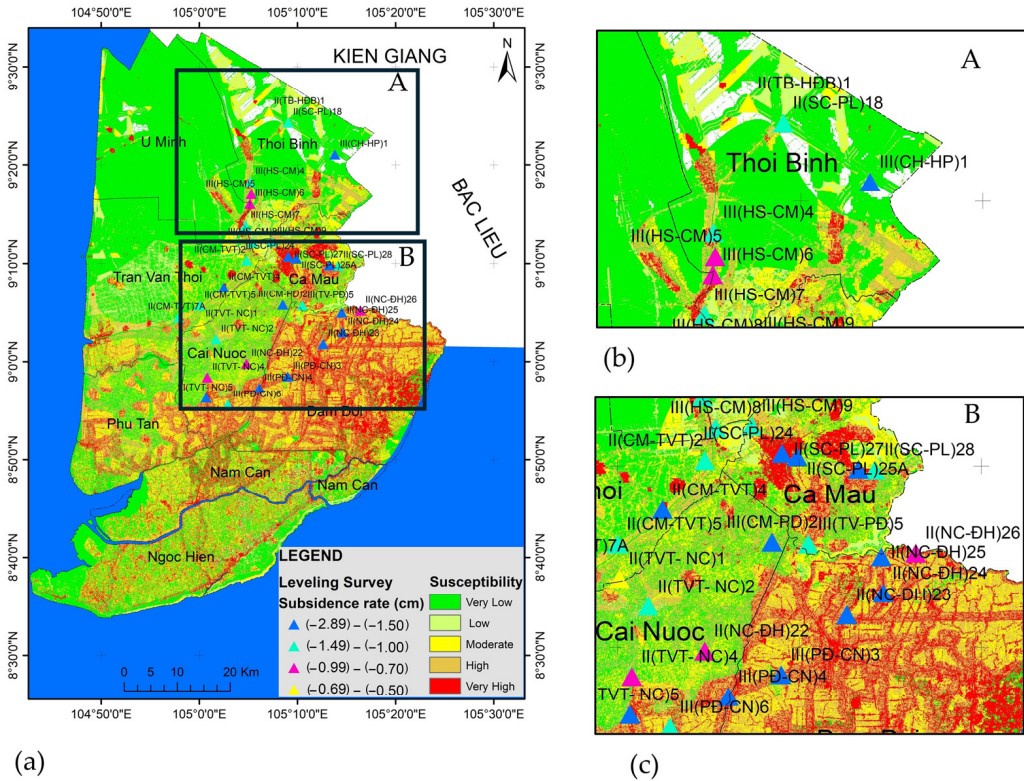

**Figure 10.** (**a**) Land subsidence susceptibility distribution map using the XGB model with leveling benchmarks; (**b**) land subsidence susceptibility distribution inside the A black rectangle; (**c**) land subsidence susceptibility distribution inside the B black rectangle.

## 6. Conclusions

The study applied boosting machine learning models, including AdaBoost (ADB), Gradient Boosting (GB), and XGBoost (XGB), to construct a subsidence susceptibility prediction map for the Ca Mau region. The inventory land subsidence data used to train and test the model were obtained using the PSI method in the EMSN062 project [17]. Additionally, eight influencing factors were considered, including the digital terrain model (DTM), land use/land cover (LULC), groundwater depth, normalized difference vegetation index (NDVI), geology, soil, distance to roads, and distance to rivers/streams. Among these input layers, LULC had the highest impact on the subsidence susceptibility prediction model, followed by groundwater depth and distance to roads. Other factors had a relatively low influence, such as DTM because the Ca Mau Peninsula has low terrain dominated by alluvial soil, making their impact less pronounced.

Among the three selected models, ADB exhibits the lowest accuracy, with the smallest AUC and ACC values observed in both the training and testing datasets compared to the other models. Between the remaining two models, GB and XGB, the accuracies are nearly equivalent; however, XGB holds a slight advantage with AUC values of 0.88 and 0.89 on the training and testing datasets, respectively. Consequently, XGB was selected as the model for predicting a subsidence susceptibility map of the Ca Mau Peninsula.

The subsidence susceptibility distribution map indicated that the highest subsidence susceptibility is in urban areas, specifically in Ca Mau City and along roads leading to the southern districts of the peninsula. The susceptibility is more closely associated with groundwater factors than other factors such as distances to rivers, geological strata, and soil composition. Groundwater depth of $qp_{2-3}$ was identified as a significant factor in building the subsidence susceptibility map. The highest subsidence occurred in areas with a groundwater depth of $-18$ m in Ca Mau City, while areas with lower subsidence susceptibility had a groundwater depth of only $-6$ m in the U Minh district. This is

demonstrated by the high correlation of three variables: LULC, groundwater depth and distance to roads with subsidence susceptibility.

The accuracy of the subsidence susceptibility prediction map generated by XGB was further validated using 31 subsidence points measured between 2005 and 2020, provided by the Department of Mapping and Geographic Information. The findings indicate a 70.9% accuracy rate in predicting subsidence compared to the leveling measurement points. This demonstrates the capability to utilize PS subsidence points derived from PSI to predict subsidence susceptibility, particularly when direct measurement methods like leveling surveys or high-precision GNSS are lacking.

In conclusion, proper land use planning and groundwater management could help address issues related to land subsidence in the Ca Mau Peninsula, contributing to sustainable economic development. By making informed decisions regarding water utilization and imposing limits on extraction, managers can mitigate the risk of land subsidence and strategize urban expansion and road construction more efficiently. Future research endeavors could focus on developing novel models and algorithms to enhance the accuracy of subsidence prediction.

**Author Contributions:** Conceptualization, Anh Van Tran and Maria Antonia Brovelli; methodology, Dong Thanh Khuc; code, Anh Van Tran and Duong Nhat Tran; validation, Maria Antonia Brovelli, Hanh Hong Tran, and Nghi Thanh Le; formal analysis, Khien Trung Ha; investigation, Khien Trung Ha; resources, Anh Van Tran; data curation, Dong Thanh Khuc; writing—original draft preparation, Anh Van Tran; writing—review and editing, Maria Antonia Brovelli; visualization, Hanh Hong Tran; supervision, Maria Antonia Brovelli. All authors have read and agreed to the published version of the manuscript.

**Funding:** This research received funding from the Scientific Research Project of the Ministry of Education and Training of Vietnam under code B2022-MDA-13.

**Data Availability Statement:** The data supporting the results of this study are available from the corresponding author upon request.

**Acknowledgments:** The authors sincerely thank the Department of Mapping and Geographic Information Systems, Vietnam Institute of Geology and Mineral Resources for providing maps for model building. We also thank the European Space Agency for providing free satellite images to create the NDVI map. In addition, we would like to thank Copernicus Emergency Management Service—Mapping for providing subsidence data in the Mekong Delta region for building a prediction model. The research team sincerely thanks the Ministry of Education and Training of Vietnam for providing financial support to project B2022-MDA-13 related to this research.

**Conflicts of Interest:** The authors declare no conflicts of interest.

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
