# Peer review of "Land Subsidence Susceptibility Mapping in Ca Mau Province, Vietnam, Using Boosting Models"

_ijgi, doi:10.3390/ijgi13050161_

Round 1
Reviewer 1 Report
Comments and Suggestions for Authors
Even though the subject of the paper seems interesting, from geotechnical point of view is average, as long it is not compared to measurements, or calculated settlements.
It is recommended to add to the research team a geotechnical engineer and then to compare some results. The fact that in the paper it is stated that “sand is often impermeable” (line 283) (and it is not!!!!) indicates the lack of geotechnical knowledge of soil behavior, therefore from geotechnical point of view, at least in this form the paper has a low scientific value.
The factors considered in evaluation are chosen without a scientific base, combining different categories of factors, without any reference to similar studies.
The algorithm of mapping seems a good one, but I suggest adding geotechnical input to it and reconsider variables and their interactions. Maybe adding some importance coefficient for each one could improve the mapping.
Comments on the Quality of English LanguageModerate editing of English language required
Reference to Figure 5 missing.
Author Response
|
1. Summary |
|
|
|
Thank you very much for taking the time to review this manuscript. Please find the detailed responses below and the corresponding revisions/corrections highlighted/in track changes in the re-submitted files
|
||
|
2. Point-by-point response to Comments and Suggestions for Authors |
||
|
Comments 1: It is recommended to add to the research team a geotechnical engineer |
||
|
Response 1: Thank you for pointing this out. We will further consider inviting a geotechnical engineer to the team for the next research focus on the Ca Mau city and surrounding areas. |
||
|
Comments 2: The fact that in the paper it is stated that “sand is often impermeable” (line 283) (and it is not!!!!) indicates the lack of geotechnical knowledge of soil behaviour, therefore from a geotechnical point of view, at least in this form the paper has a low scientific value |
||
|
Response 2: We have revised. We had some confusion while writing this paragraph. We have corrected it at line 342 Comments 3: The factors considered in the evaluation are chosen without a scientific base, combining different categories of factors, without any reference to similar studies. Response 3: We have revised. The impact factors have been referred to several references and further explained at line 304 Comments 4: The algorithm of mapping seems a good one, but I suggest adding geotechnical input to it and reconsider variables and their interactions. Maybe adding some importance coefficient for each one could improve the mapping. Response 4:. Thank you for your suggestion. As analyzed in the question above, we have consulted articles related to building subsidence susceptibility prediction models and consulted experts to choose input factors for this model at line 304 Comments 5: Reference to Figure 5 missing. Response 5:. Thank you for your comment. We have revised at line 492
|
||
|
4. Response to Comments on the Quality of English Language |
||
|
Point 1: Moderate editing of English language required |
||
|
Response 1: I have reviewed and revised the English language |
||
|
|
||

Reviewer 2 Report
Comments and Suggestions for Authors
Minor Revision
Good paper. Well written with a good organization.
References good.
Figures presented are mostly ok but you need more: All the "Influence Factors" maps need to be presented in Figures and references listed in the Reference list. Also Figure 1 needs a better caption, Figure 4 and Figure 5 need higher resolution, and Figure 9 needs color legends.
A marked up manuscript with 43 comments/suggestions is attached.
Detailed Comments
1) Line 68-69 - Even though the AdaBoost was 81.1%, the other algorithms were also quite close (79.1%-80%).
2) Figure 1 - The Figure 1 caption needs to indicate what is shown in the figure including the points on the right side map.
3) Line 178 - You have variable "xi". Should not the "i" be a subscript as in the equation?
4) Line 184 - Looks like you are showing an Greek alpha letter instead of an X. There are no alpha's in Eqtn 1.
5) Figure 2 - Should all those alpha's be x's?
6) Line 188 - Shouldn't this be referring to Figure 2 instead of Figure 1?
7) Line 199 - Why is this good: "GB tries to build a decision tree to match the resid-199 uals from the previous model"?
8) Line 211 - You need to cite a reference for XGB: "mainly developed by Tianqi Chen"
9) Line 275 - Show a figure with this map and provide a reference with a URL link: "geological map of Ca Mau"
10) Line 285 - Show a figure with this map and provide a reference with a URL link: "Soil map of Ca Mau"
11) Line 292 - Show a figure with this map and provide a reference with a URL link: "LULC map of Ca Mau"
12) Line 311 - Show a figure with this map and provide a reference with a URL link: "NDVI map of Ca Mau area is made from Sen-311 tinel-2 satellite images"
13) Line 320 - Show a figure with this map and provide a reference with a URL link: "The road map"
14) Line 329 - Show a figure with this map and provide a reference with a URL link: "The river map"
15) Table 1 - Provide a legend for the geology units abbreviations.
16) Table 1 - Do you really need 3 decimal places in the Table?
17) Figure 4 needs to be of much higher resolution. Can't read the maps or legends.
18) Figure 5 - Figure 5 also needs to be of much higher resolution. Can't read the details.
19) Lines 389-390 - The English is not clear on these lines. Rephrase.
20) Figure 8 looks good.
21) Figure 9 - Figure 9 needs a color legend indicating what blue means and what red means in Fig 9a and what green means in Fig 9b and what green and red mean in Figure 9c.

Included in comments above and in attached reviewed pdf
Author Response
|
1. Summary |
|
|
|
Thank you very much for taking the time to review this manuscript. Please find the detailed responses below and the corresponding revisions/corrections highlighted/in track changes in the re-submitted files
|
||
|
2. Point-by-point response to Comments and Suggestions for Authors |
||
|
Comments 1: Figures presented are mostly ok but you need more: All the "Influence Factors" maps need to be presented in Figures and references listed in the Reference list |
||
|
Response 1: Thank you for pointing this out. We have revised based on the reviewer's comments at at line 314, 321, 328, 336, 351, 360, 366, 386, 406 |
||
|
Comments 2: Figure 1 needs a better caption |
||
|
Response 2: We have revised the map legend, adding more description in the figure caption at line 146-147 Comments 3: Figure 4 and Figure 5 need higher resolution Response 3: We have,revised at line 483-487, 491 Comments 4: Figure 9 needs color legends.. Response 4:. We have revised based on the reviewer's comment at line 660 Comments 5: Line 68-69 - Even though the AdaBoost was 81.1%, the other algorithms were also quite close (79.1%-80%). Response 5:. Thank you for your comment. We have,revised at line 80-82 Comments 6 Figure 1 - The Figure 1 caption needs to indicate what is shown in the figure including the points on the right side map. Comments 7: Line 178 - You have variable "xi". Should not the "i" be a subscript as in the equation? Comments 8: Line 184 - Looks like you are showing an Greek alpha letter instead of an X. There are no alpha's in Eqtn 1 Comments 9: Figure 2 - Should all those alpha's be x's? The number of input variables x and the number of alpha weights are not the same because the number of weights depends on the number of trees that are selected Comments 10: Line 188 - Shouldn't this be referring to Figure 2 instead of Figure 1? We were spelled wrong. We have revised it at line 214 Comments 11: Line 199 - Why is this good: "GB tries to build a decision tree to match the residuals from the previous model"? I'm just making a distinction between the two algorithms, I don't mention that GB is better than AdaBoost. I have explained more in the paper Comments 12: Line 211 - You need to cite a reference for XGB: "mainly developed by Tianqi Chen" Comments 13: Line 275 - Show a figure with this map and provide a reference with a URL link: "geological map of Ca Mau" Comments 13: Line 285 - Show a figure with this map and provide a reference with a URL link: "Soil map of Ca Mau" Comments 14: Line 311 - Show a figure with this map and provide a reference with a URL link: "NDVI map of Ca Mau area is made from Sentinel -2 satellite images" Comments 15: Line 320 - Show a figure with this map and provide a reference with a URL link: "The road map" 418 Comments 16: Line 329 - Show a figure with this map and provide a reference with a URL link: "The river map" 434 Comments 17: Provide a legend for the geology units abbreviations. [After re-examination, we considered re-dividing the stratigraphic layers of the geological map to be more reasonable because some types are located on islands near the shore without data to calculate FR values. Details have been annotated below the figure.] Comments 18: Table 1 - Do you really need 3 decimal places in the Table? Comments 19: Figure 4 needs to be of much higher resolution. Can't read the maps or legends. Comments 20: Figure 5 - Figure 5 also needs to be of much higher resolution. Can't read the details. Comments 20: Lines 389-390 - The English is not clear on these lines. Rephrase. Comments 21: Figure 9 - Figure 9 needs a color legend indicating what blue means and what red means in Fig 9a and what green means in Fig 9b and what green and red mean in Figure 9c.
|
||
|
4. Response to Comments on the Quality of English Language |
||
|
Point 1: Moderate editing of English language required |
||
|
Response 1: I have reviewed and revised the English language |
||
|
|
||
|
|
||

Reviewer 3 Report
Comments and Suggestions for Authors
Please see attached paper.

Author Response
|
1. Summary |
|
|
|
Thank you very much for taking the time to review this manuscript. Please find the detailed responses below and the corresponding revisions/corrections highlighted/in track changes in the re-submitted files |
||
|
|
|
|
|
3. Point-by-point response to Comments and Suggestions for Authors |
||
|
Comments 1: [Line 25. Here in the abstract you mention that the sample is split into 70% training and 30% testing points. This is not mentioned anymore in the methods part. Please integrate it there too..] Response 1: [I have added it as comment at 491, 501] |
||
|
Comments 2: [Line 34-36. I guess it would make sense to mention that GS is also a natural process, even if no anthropogenic influences are around.] Response 2: Yes, I find this idea very accurate. I have edited according to your comments |
||
|
Comments 3: [Line 133-142. The height of the land relative to the sea level is time dependent as sea level rise and GS together result in effective sea level rise (SLR). This changes effective land heights over time. I suggest that you state the source of your data and for what year they apply. In this context it might be important to note that SRTM data are now more than 20 years old and with the observed GS rates in the Mekong Delta plus climate change induced SLR you may have something like 50 cm lower land levels (base: sea level) than around the year 2000.. Response 3: I have revised at line 319-321 [The SRTM DEM is so far compared to now. We didn't pay attention to this. Instead, we utilized the elevation layer derived from the 1:50000 scale topographic map in 2010 provided by the Department of Survey, Mapping and Geographic Information Vietnam to enhance the accuracy of the model construction.] |
||
|
Comments 4: [Line 148 – 159. Please indicate the sources for your statements. If there is data available concerning the number, size and abstraction quantities they should be mentioned here. You may consider moving this topic to Chapter 4 and using the respective data as an additional factor that may influence GS. It might be worthwhile to have a look at the data in this publication: NAWAPI, BGR: Technical Note TN-IV-05, Groundwater abstraction maps for the Mekong Delta in 2017, published 2021, Appendix 3 might be helpful as these data reflect ground water abstraction per sqkm.] Response 4: [I have edited according to the comments to move the information in lines 148-159 to section 4. Reference information has been added. I have read NAWAPI's document, BGR: Technical Note TN-IV-05, Groundwater abstraction maps for the Mekong Delta in 2017, published 2021 and have consulted some information and included it in the article. However, the data I referenced from NAWAPI 2022 has data in the form of an Excel table, so I still use this data because the report does not have a specific data table.] |
||
|
Comments 5: [Please check the small/big caps in “Adaboost” in the paper. It is inconsistent in the paper.] Response 5: [Yes, I have checked and corrected it at line 81] |
||
|
Comments 6: [Line 253 – 259. There is a need to describe how you produced your InSAR points. There are different methods available, and they may result in slightly different outputs. You used 1910 InSAR data points for the province. Why were only 1910 PS selected? What were the selection criteria? Please note that there are 48,400 PS points in the data set of Copernicus Response 6: [Subsidence data is taken from subsidence points established using the PSI method but we did not process it. We referred to the data at https://emergency.copernicus.eu/mapping/list-of-comComponents/EMSN062. referenced in document number 14 (old version). After checking the evidence that the PSI processing results of document number 14(old version). are good and referenced in the author's article 15(old version)., I boldly used this data as input data for the model. Two documents are 17 and 18 in a new version at line 291, 292. There are many PS subsidence points, but we have filtered out the points with subsidence greater than -1cm/year. After filtering, only 1011 points with subsidence were selected. Besides, for the model given training data, there must be two values: subsidence and non-subsidence, so we have to choose an additional point without subsidence. These points collect PS points with approximate values in the range close to 0. After redoing the point data, we have rebalanced the subsidence and non- subsidence points almost equally. The total number of settlement and non-settlement points included in the model is now 2011 points (old dataset was 1910 points).] |
||
|
Comments 7: [Line 264 – 267. Can you give a source for your statement?] Response 7: [We have revised according to comments at line 314] |
||
|
Comments 8: [Line 271. You took the SRTM 30m DEM to represent the topography of Ca Mau. You may notice that there is considerable noise in this dataset. You have pixels lying more than 7m below sea level. Obviously, this would be just water if it is true. A look at Google Earth would reveal that such pixels are not water but land, I think. Two questions in this context: Why did you chose SRTM and how did you deal with obvious outliers? You may also consider the following DEMs or give a reason why you prefer a particular DEM:] Response 8: We changed the SRTM DEM to another at line 318-320 [We were really negligent in choosing the 30m SRTM DEM with some noise, particularly when negative values reach up to -7m. Additionally, we realize that this DEM is outdated, as it was generated in 2000, which significantly deviates from the present time. Following your suggestion, we attempted to obtain a DEM based on the Vietnamese topographic map through interpolation, but it had insufficient resolution at 500m. We also considered the Merit DEM sourced from NASA's SRTM DEM with a resolution of 3" (90m), as well as the TanDEM-X with a resolution of 90m, both of which have a lower precision. Ultimately, we opted to replace the data with a DEM extracted from a 1:50,000 scale topographic map provided by the Department of Mapping and Geographic Information as mentioned above.] |
||
|
Comments 9: [Line 272 -276. You may consider the age of land in Ca Mau as a factor. The age of land in Ca Mau City is approximately 3000 years old while the southern tip of the province is less than 1000 years old. This might be introduced as a factor for investigation based on the assumption that younger land is still more compacting than older land. Source: Fig 6 in: Liu, J.P., D.J. DeMaster, T.T. Nguyen, Y. Saito, V.L. Nguyen, T.K.O. Ta, and X. Li. 2017. Stratigraphic formation of the Mekong River Delta and its recent shoreline changes. Oceanography 30(3):72–83, https://doi.org/10.5670/oceanog.2017.316.] Response 9: [Thank you for referencing this document and I included part of the article's information in our article at line 328.] |
||
|
Comments 10: [You may notice considerable differences between these maps for Ca Mau. Please discuss why you chose a particular source and not one of the other ones. Depending on the map you use, the results of the calculations may change considerably.] Response 10: I revised at line 337-351 [In Vietnam, when choosing a map, the Center of Survey and Mapping Data, Department of Survey, Mapping and Geographic Information is the place to store and update maps of the whole country. In addition, the maps here are guaranteed for accuracy and reliability when used. If this center does not have it, we will choose another place to provide it. Online maps are often rarely verified, so we limit their use. For soil maps, the map scale is 1:100,000 and soil types are divided into smaller parts. I found the suggestion to refer to some soil maps to be reasonable, so I re-selected a larger soil group to determine the input factors of the subsidence susceptibility model. I have now redone this land map in 4 layers.] |
||
|
Comments 11: [Line 282 – 284. You may consider changing the sentence to: “Even though clay is almost impermeable, it can hold large amounts of water and swell when water permeates and may hence release the water and compress much more under high pressure than sand layers.] Response 11: [[We have revised according to comments at line 339-344 ] |
||
|
Comments 12: [Line 287 – 293. There are different sources publishing LU/LC maps (e.g. Copernicus, FAO). Please state why you selected the one from the Dep. Of Survey, Mapping and Geographic Information. LU/LC is changing over time. Please indicate what year the used data refers to.] Response 12: [We mislabeled the supply source. Our LULC map is provided by the Department of Natural Resources and Environment of Ca Mau province. The map of current land use status was made in 2014. In the Ca Mau area, we have contacted the Department of Natural Resources and Environment for the latest updated map, but currently only the 2014 map is available. The maps of FAO is a global map, so the accuracy assessment for these maps has not been proven to be guaranteed] |
||
|
Comments 13: [Line 306 – 312. There are also different NDVI maps available. NDVI is also very dependent on season and the specific year (e.g. drought). Can you explain why you chose Sentinel 2 and for what date/period?] Response 13: I added more information at line 378-383 [I captured the Sentinel-2 images for NDVI computation in 2021 from July to August. This timeframe coincides with the onset of rice planting, allowing us to differentiate between various types of vegetation, including forests, and paddy fields] |
||
|
Comments 14: [Line 313 – 322. Like the other factors, roads are also time dependent as new roads are built frequently. Please indicate what year your data is based on. The road map in Fig 4 appears to show road fragments without connection to the main road network. My PDF file does not have a good resolution for this map and I might not see some details. Did you consider using other sources like Open Street Map? Why do you take the one from Dep. Of Survey, Mapping and Geographic Information and not another source?] Response 14: I added more information at line 418-421 [Thank you for your suggestion. Our map is from the Department of Mapping which is a reliable source, however the data is a bit old and many small routes are not shown on this map. As suggested, I took the traffic map from Open Street map (OSM). However, because OSM has too many small roads, even rice fields, so in this study we took road data from OSM of the tertiary road type to update the road layer from the topographic map we created. I have used it before.] |
||
|
Comments 15: [Line 333 – 340. Ground water is the subject of many researches and you find a number of publications showing different aquifers. It would be important to know what you exactly refer to. Furthermore, does your data reflect a temporal average? The map is (I presume) based on measurements in some points in Ca Mau and you or NAWAPI did some interpolation with these point data, and you arrived at a map showing ground water levels for the whole area of the province. How many points were used, how was interpolated? There is another aspect. Ground water levels are decreasing over the years (Line 489) and if there is an influence of this dynamic on ground subsidence you would assume that GS is accelerating. This means to detect such a correlation you would have to use time-series of GS data. I don’t see this in your analysis. It might be helpful to check the timeseries of the InSAR-based GS data to find points with acceleration. They should correspond to wells with big amounts of water abstraction, if there is indeed correlation with ground water abstraction. BTW. You state that you used data from 2022 (line 338) while you quote a source (reference 19) that covers data for 2021 (a yearbook). Can you check this, please?] Response 15: I added more information at line 384-406 [For groundwater data, I used qp2-3 layer data averaged over a period of 3 years: 2019-2021. For wells in other levels we do not have enough data. Additional, explanatory information has been included in the information paragraph for groundwater data. The data points provided were interpolated by the research team using the Kriging method. The number of wells is 6 points but mainly concentrated in 3 points: Ca Mau City (2 points), Cai Nuoc, Nam Can, Thoi Binh, U Minh 1 point each. Regarding the statement in line 338, I have revised it to match the reference.] |
||
|
Comments 16: [Line 341. Data standardization chapter. You write that you used 1950 points in this chapter. This is 1910 from InSAR and 40 from geodetic surveys? Please clarify.] Response 16: [As stated earlier, I exclusively utilized subsidence data points obtained from PSI. The additional 40 points derived from geodetic surveys were employed solely for result validation purposes. As detailed in the data section, when constructing a model to predict subsidence susceptibility, two types of data are required: subsidence and non-subsidence. The 40 subsidence survey points provided by the Mapping Department served as validation data. However, in the updated version, upon reviewing the geodetic survey data, it was found that some points were incomplete values. Consequently, the new version of geodetic surveys comprises only 31 points] |
||
|
Comments 17: [Line 348. You may want to replace “formation” with “occurrence] Response 17: [I have revised according to your comments at line 466] |
||
|
Comments 18: [Can you put the four parameters N into the Table 1 and name them accordingly? This would make it easier to understand how you calculated the FR values.] Response 18: [Yes, I added in Table 1 at line 477] |
||
|
Comments 19: [Npix 3 is the number of subsidence pixels. Can you explain how this is determined? Is there a threshold? Is strong and weak GS considered or carries small/big subsidence the same weight?] Response 19: [Yes, that's right. Npix 3 is the total number of subsidence points selected to be included in the model to calculate the influence of subsidence on each sub-class in the affected layers. The subsidence points, after being taken from the PSI points, will be filtered to select points with high subsidence greater than -1cm/year. Then, dense points close together with the same value will be filtered again for reduction. As a result, we have 1011 subsidence points left. . So Npix3 will be 1011. When building the land subsidence susceptibility model, we only choose two variables: yes and no subsidence. Therefore, besides the subsidence points mentioned above, it is necessary to select an approximately equivalent number of non-subsidence points. That's why you see that the number of points included in the model is 1950 points. This time we revised our manuscript, and more non-subsidence were selected to add more for balance and now the number of points increased is 2011 points. Non-subsidence points are taken from PSI values with a threshold of less than -0.2cm/year. Because the data entered into the model is binary, with subsidence (1) and non-subsidence (0), no weights are selected in this training layer, but the weights are assigned when entering the prediction model. I want to explain more about the 8 factors that affect the subsidence of the area, which are the 8 input data layers. Each class when included is not in the same unit. For example, altitude varies in meters and LULC is labeled 1,2,3,4,5,6. They need to be standardized to evaluate the influence of each layer. Thus the FR formula is used. Formula (4) calculates the FR value, but in table 1, we have written simplified columns to make it easier to understand, calculating the % of subsidence of each subclass and the % of the class compared to the sum of all pixels in the study area. Now to calculate FR, just take the %Land subsidence column divided by %class Pixels. If formula 4 is used, you will have the same result (Npx3 takes the total of the Ls points column and Npx4 is the total of the Class pixels column).] |
||
|
Comments 20: [Is there a threshold? Is strong and weak GS considered or carries small/big subsidence the same weight?] Response 20: [For PSI data points, there is really uncertainty about the accuracy of the subsidence values, so in this study we only focus on getting the distribution of subsidence points on the influence layers instead because we did not take the subsidence value to train the model. Therefore, we had to use the FR formula to calculate the influence of concentrated subsidence points on each subclass.] |
||
|
Comments 21: [Concerning the normalization I suggest that you have a look at your data and do not consider outliers not making sense like the -7 m altitude of the DEM] Response 21: [Yes I have seriously corrected it. The DEM was obtained from the interpolated topographic map. Therefore, all model data will be re-run from the beginning.] |
||
|
Comments 22: [Can you explain what the LS points in Table 1 are reflecting? You have 1950 points in total, but the points listed here for the different factors are summing up to 1032 (Elevation) or 1026 (Soil) etc.. You may want to rename “Sub-factor” to “Class”. This would be consistent with the other columns in the Table 1.] Response 22: [Yes as explained above. I have included in the calculation the number of subsidence points as 1011 points. The total score for both subsidence and non-subsidence is 1950 (previous data). Currently, I have reprocessed the entire thing and now the total subsidence and non-subsidence points are 2011 points including 1011 subsidence points and 1000 non-subsidence points. I changed Sub-factor” to “Class”.] |
||
|
Comments 23: [Please check the use of different bracket styles in the column Sub-factor or class. This appears to be inconsistent.] Response 23: [I checked and revised it at line 477-478] |
||
|
Comments 24: [Please check the numbers in the column “% Land subsidence”. For some factors you have a total of 100.01 %.] Response 24: [Yes, I checked and revised it at line 569,570] |
||
|
Comments 25: [In Table 1 you have 22 GS points in water (LULC factor). Please consider omitting this as neither leveling nor InSAR data were collected in water. Furthermore, in the LULC category you have six classes and you put them into a particular order. Is there evidence suggesting that order? Please explain.] Response 25: [Yes, I checked and revised it at line 483 LULC category does not require any order, it can be arranged in any order because after we use FR to calculate new values for sub-classes, there is no order for these data anymore.] |
||
|
Comments 26: [Chapter 5. I am missing results concerning the proximity of waterways and roads. You worked with different buffers, and I expected results saying something about the susceptibility close or far from these features. Can you include this?] Response 26: [I have revised and added analyzes related to roads and hydrology.] |
||
|
Comments 27: [Line 427 – 428. In the upper part of Table 3 a dividing line appears to be missing. Response 27: [Yes, I checked and revised it at line 569, 570. I confused the results of the previous calculation when using the data set divided by 80% and 20% for training and testing] |
||
|
Comments 28: [Line 456 – 457. Here, it would be very interesting to see a comparison of a map with actual GS with your map showing GS susceptibility. If you take the data from Copernicus (EMSN062) with an interpolated map you will see patterns that are quite different from your GS susceptibility map (Fig. 8a).] Response 28: I revised at lines 664-710 [Regarding the subsidence susceptibility map, it is related to many factors, while the subsidence monitoring map using PSI monitors subsidence over a period of time, so it may be a little bit different, but it cannot be said that there is quite difference as long as those points are not overlapped. I have incorporated 31 points from the levelling survey onto the land subsidence susceptibility map for validation, demonstrating satisfactory accuracy.] |
||

Reviewer 4 Report
Comments and Suggestions for Authors
Dear Authors,
I have read your manuscript carefully and, firstly, I wish to congratulate you on your interesting research. I have no doubts that such automatic classification and prediction relying on machine learning techniques is awaited in the cartographic world. I assess your text positively and am convinced it should be published. Nevertheless, before this happens, I suggest you undergo an in-depth check, resulting in some necessary improvements.
First, please check the text in a technical sense - correct missing spaces or lines, different font sizes (for example, in line 4), and so on. Although your article is understandably and correctly written, I recommend you proofread it again.
Secondly, while reading your paper, I spotted some incorrectness needing improvement:
- line 18: a word starting with the capital letter ("Boosting"),
- the ROC acronym needs to be explained before being used in the text,
- line 44: mentioning Industry 4.0 needs to be explained and justified (except for the fact that it is using AI techniques); what is more, I suggest citing a relevant position discussing that,
- in the "Introduction" you mention different studies relevant to your research, which is correct, but please expose your novelty more clearly. From the current text, one can find out that the only reason you have been doing your tests is to check one algorithm (lines 115-123),
- Figure 1 - the right picture (map) is hard to read, especially the scale bar is far too small - so are the remaining names. The descriptions in the legend are practically unreadable.
- lines 143-147: providing such data needs references,
- line 155: like above,
- line 166: I recommend to cite the relevant references confirming that fact,
- formula 1 and figure 2: both need references or information about copyrights,
- line 188: you probably thought about figure 2,
- formula 2: references needed,
- figure 3 needs either references or information about copyrights,
- please use a unified form of providing the formulas - they all should be written in the equation editor (in your text, it is differently written),
- figure 4 - too many small maps make them completely unreadable; their elements are far too small. The legends contain some tiny descriptions that are impossible to read. There are some diagrams, but what do they really mean? I recommend you change how they are presented by either presenting each thematic map separately or enlarging them respectively. By the way - while looking carefully at the charts, I can see improper color scales, but bigger pictures are mandatory to tell you more details!
- figure 5 is, unfortunately, unacceptable! Please see above to get more information,
- figure 6: each axis must be explained and completed by units; moreover, please either provide references or information about copyrights,
- formulas 5,6,7 differ from the previous ones, so - please unify the way of presenting them,
- figure 7 is too small; it is very challenging to get information based on it,
- figure 8: the scale bar must be enlarged,
- figure 9: the cartographic grid is far too small to be read; the map lacks color scale, although it presents hypsometric form.
Should your text be accordingly corrected, please submit it for another review cycle. Good luck!
Comments on the Quality of English Language
The English language is generally correct, but I recommend undergoing a thorough proofreading, hence there are some mistakes and technical errors.
Author Response
|
1. Summary |
|
|
|
Thank you very much for taking the time to review this manuscript. Please find the detailed responses below and the corresponding revisions/corrections highlighted/in track changes in the re-submitted files. |
||
|
|
|
|
|
3. Point-by-point response to Comments and Suggestions for Authors
Comments 1: [First, please check the text in a technical sense - correct missing spaces or lines, different font sizes (for example, in line 4), and so on. Although your article is understandably and correctly written, I recommend you proofread it again.] Response 1: [Yes, I checked and revised it ] Comments 2: [- line 18: a word starting with the capital letter (""Boosting""),] Response 2: [Yes, I checked and revised it at line 19 ]
Comments 3: [- the ROC acronym needs to be explained before being used in the text,] Response 3: [I checked and revised it at line 28 ]
Comments 4: [- line 44: mentioning Industry 4.0 needs to be explained and justified (except for the fact that it is using AI techniques); what is more, I suggest citing a relevant position discussing that,] Response 4: [This is just a sentence written by my opinion. I find your comments very correct. I have edited and omitted this sentence at line 52.]
Comments 5: [- in the ""Introduction"" you mention different studies relevant to your research, which is correct, but please expose your novelty more clearly. From the current text, one can find out that the only reason you have been doing your tests is to check one algorithm (lines 115-123)] Response 5: [In the section stating the reasons for choosing the research we wrote in 2 paragraphs, I explained the lack of research related to subsidence prediction, but the research here mainly focuses on subsidence monitoring. I have also added several additional paragraphs for further explanation regarding low terrain and the need for managers and planners to properly use land in light of this research. Some adding in line 115-135] Comments 6: [- Figure 1 - the right picture (map) is hard to read, especially the scale bar is far too small - so are the remaining names. The descriptions in the legend are practically unreadable. ] Response 6: [I have revised this Figure 1 at line 139] Comments 7: [- lines 143-147: providing such data needs references,] Response 7: [I have added references at line 162]
Comments 8: [- line 155: like above,] Response 8: [According to reviewer number 2's comments, I moved this section to section 4 and added references. at line 337-351]
Comments 9: [- line 166: I recommend to cite the relevant references confirming that fact,] Response 9: [I have corrected the sentence and included references at line 199]
Comments 10: [- formula 1 and figure 2: both need references or information about copyrights,] Response 10: [I have included reference at line 207,208] Comments 11: [- line 188: you probably thought about figure 2,] Response 11: [That's right, I got confused here, this is the caption pointing to Figure 2 at line 216]
Comments 12: [- formula 2: references needed,] Response 12: [I have included reference]
Comments 13: [- figure 3 needs either references or information about copyrights,] Response 13: [I have included reference at line 219]
Comments 14: [- please use a unified form of providing the formulas - they all should be written in the equation editor (in your text, it is differently written),] Response 14: [I have revised the formulas as suggested at line 367,472] Comments 15: [- figure 4 - too many small maps make them completely unreadable; their elements are far too small. The legends contain some tiny descriptions that are impossible to read. There are some diagrams, but what do they really mean? I recommend you change how they are presented by either presenting each thematic map separately or enlarging them respectively. By the way - while looking carefully at the charts, I can see improper color scales, but bigger pictures are mandatory to tell you more details!] Response 15: [I revised it to make it clearer and increased the image resolution to 600dpi] Comments 16: [- figure 5 is, unfortunately, unacceptable! Please see above to get more information,] Response 16: [I have remade the Flow chart in figure 5 in a different way to make it easier to understand and see at line 491] Comments 17: [- figure 6: each axis must be explained and completed by units; moreover, please either provide references or information about copyrights,] Response 17: [I have edited and added notes on the coordinate axes at line 530. This figure shows the importance of 8 data layers affecting the susceptibility of land subsidence.] Comments 18: [- formulas 5,6,7 differ from the previous ones, so - please unify the way of presenting them,] Response 18: [I have revised these three formulas into the correct format] Comments 19: [- figure 7 is too small; it is very challenging to get information based on it,] Response 19: [I adjusted the resolution and made the image larger so it's easier to see] Comments 20: [- figure 8: the scale bar must be enlarged,] Response 20: [I have revised it according to your comment at line 615] Comments 21: [- figure 9: the cartographic grid is far too small to be read; the map lacks color scale, although it presents hypsometric form.] Response 21: [I have revised it according to the reviewer's comments. Added color palette to the Figure 9 at line 660]
|
||

Round 2
Reviewer 1 Report
Comments and Suggestions for Authors
Thank you for considering the comments and improving the paper.
Reviewer 4 Report
Comments and Suggestions for Authors
Dear Authors,
Thank you for considering my suggestions and submitting a revised version of your manuscript. I appreciate all your improvements and think your text is ready for publishing after the editors require all the remaining necessary checks.
I wish to congratulate you on your interesting studies. Good luck!